Journal of Data-centric Machine Learning Research (2024)        Submitted 8/24; Revised 11/24; Published 11/24

# Evaluating Durability: Benchmark Insights into Image and Text Watermarking

**Jielin Qiu**[1,2*†]                                      JIELINQIU@GOOGLE.COM
**William Han**[2*]                                        WJHAN@ANDREW.CMU.EDU
**Xuandong Zhao**[3]                                  XUANDONGZHAO@BERKELEY.EDU
**Shangbang Long**[1]                                 LONGSHANGBANG@GOOGLE.COM
**Christos Faloutsos**[2]                                 CHRISTOS@CS.CMU.EDU
**Lei Li**[2]                                                    LEILI@CS.CMU.EDU
[1] *Google DeepMind*
[2] *Carnegie Mellon University*
[3] *University of California, Berkeley*

**Reviewed on OpenReview:** *https://openreview.net/forum?id=IaUzVUf0CL*

**Editor:** Hongyang Zhang

## Abstract

As large models become increasingly prevalent, watermarking has emerged as a crucial technology for copyright protection, authenticity verification, and content tracking. The rise of multimodal applications further amplifies the importance of effective watermarking techniques. While watermark robustness is critical for real-world deployment, the current understanding of watermark robustness against various forms of corruption remains limited. Our study evaluates watermark robustness in both image and text domains, testing against an extensive set of 100 image perturbations and 63 text perturbations. The results reveal significant vulnerabilities in contemporary watermarking approaches - detection accuracy deteriorates by more than 50% under common perturbations, highlighting a critical gap between current capabilities and practical requirements. These findings emphasize the urgent need for more robust watermarking methods that can withstand real-world disturbances. Our project website can be found at `https://mmwatermark-robustness.github.io/`.

**Keywords:** image and text watermarking, robustness, image corruptions and text perturbations, multimodal

## 1 Introduction

Watermarks represent a sophisticated method of embedding information within digital content across various modes, such as audio, video, and images. These watermarks are designed to be imperceptible, or nearly so, to the human senses, yet detectable by specialized software or algorithms. The primary purpose of a watermark is to assert copyright or verify the authenticity of the content. Unlike traditional watermarks, which are limited to a single type of media, watermarks can be integrated across different formats, enhancing security and

---

*. Marked as equal contribution.

†. Work done while at Google.

flexibility. The robustness of these watermarks against various forms of perturbations and their ability to remain intact even when the content is transformed or compressed is a key aspect of their design, making them essential in generated content identification, copyright protection, and digital rights management.

The robustness of watermarks under various perturbations is a critical aspect of their effectiveness and reliability. In the digital realm, image content often undergoes a variety of transformations, such as compression, scaling, cropping, or format conversion, while textual content often undergoes synonym replacement, paraphrasing, or typing differences, which can potentially alter or obliterate embedded watermarks. The robustness of a watermark to withstand these perturbations is vital to ensure that the embedded data remains intact and retrievable. This is particularly important for copyright protection, piracy detection, and the verification of the authenticity of digital media. A robust watermark ensures that ownership rights are preserved and content integrity is maintained, even when the media is shared across different platforms and undergoes multiple alterations. Furthermore, in sensitive applications such as legal documentation or secure communications, the persistence of a watermark through various distortions is crucial for maintaining the trust and reliability of the information contained within the media. Therefore, understanding the watermark's robustness to perturbations is a key focus.

To our best knowledge, there is currently no comprehensive study of how the perturbations can affect the performance of image and text watermarks. Hence, in this work:

- We evaluate watermark robustness under image and text perturbations by analyzing 4 image watermarking methods and 4 text watermarking methods. Our study tested the performance of 8 image-to-text models and 8 text-to-image models against 100 image perturbations and 63 text perturbation methods.

- We find that watermarks are sensitive to distribution shifts caused by image and text perturbations. Specifically, for image perturbations, Zoom Blur consistently shows the highest impact, while Glass Blur is the least harmful one. For text perturbations, Casual consistently shows the highest impact, while OCR is the least harmful. Under image perturbations, SSL-WM seems more stable; while under text perturbations, KGW-WM seems more stable.

- We have publicly released our codebase at `https://mmwatermark-robustness.github.io/` with CC BY-NC-SA License.

## 2 Related Work

**Text Watermarking** has become increasingly relevant due to the usage of language models (LMs) for text generation (Liu et al., 2024). Text watermarking frameworks should integrate seamlessly into a model with minimal impact on the generated text, without altering the model's parameters (Kirchenbauer et al., 2023b; Aaronson, 2023; Christ et al., 2023). Additionally, text watermarks must be robust against distribution shifts, leading to the proposal of a watermarking algorithm that assigns a sequence of arbitrary numbers generated by a random watermark key to a sample from the LM (Kuditipudi et al., 2023). However, relying solely on empirical methods to assess the effectiveness of proposed watermarking algorithms is insufficient. In response, a theoretical framework to quantify the robustness

of text watermarks was introduced. From this theoretical analysis, an enhanced framework that utilizes a fixed grouping strategy was proposed (Zhao et al., 2023a). However, these works lack comprehensive evaluations of their watermarking systems under a variety of perturbations.

**Image Watermarking** studies protecting intellectual image property (Cox et al., 2007). Recently, encoder/decoder models have been introduced (Ahmadi et al., 2020; Lee et al., 2020; Luo et al., 2020; Zhang et al., 2020a; Zhu et al., 2018; Fernandez et al., 2022a; Kishore et al., 2022; Vukotić et al., 2018), which have shown promising results in terms of robustness against a broad array of transformations. In the realm of generative models, there have been attempts to watermark the training datasets used to train these models (Yu et al., 2021), an approach that is markedly inefficient as embedding each new message necessitates a separate training pipeline. A more contemporary strategy involves integrating the watermarking process directly with the generative process (Fei et al., 2022; Lin et al., 2022; Nie et al., 2023; Qiao et al., 2023; Wu et al., 2020; Yu et al., 2022; Zhang et al., 2020b), aligning it more closely with the broader literature on model watermarking (Uchida et al., 2017).

**Robustness of Image or Text Watermarks** An et al. (2024) examined the vulnerabilities in various image watermarking techniques. Zhao et al. (2023b) and Saberi et al. (2023) found that methods like noising and denoising through diffusion models can effectively remove some watermarks. Jiang et al. (2023) studied the robustness of AI-generated content detection that relies on watermarking. Mofayezi and Medghalchi (2023) explored the impact of text-guided corruptions on image classifiers. Kirchenbauer et al. (2023c) optimized the watermark generation and detection pipeline for greater reliability in real-world scenarios. Our study aims to challenge both text and image watermarking systems using perturbations to identify potential vulnerabilities and robustness.

**Robustness of Multimodal Models** For the robustness evaluation of multimodal image-text models under distribution shift, previous works (Goh et al., 2021; Daras and Dimakis, 2022; Galindo and Faria, 2021; Fort, 2021; Goh et al., 2021; Noever and Noever, 2021) have tested some pre-trained models, such as CLIP (Radford et al., 2021), by attacking them with text patches and adversarial pixel perturbations. Notably, Daras and Dimakis (2022) discovered that DALLE-2 (Ramesh et al., 2022) possesses a hidden vocabulary that enables image generation from absurd prompts, questioning the robustness of its output. Fang et al. (2022) attributed robustness gains primarily to diverse training distributions. Cho et al. (2022) explored the robustness of text-to-image generative models concerning visual reasoning capabilities and social biases. For benchmarking robustness, (Li et al., 2021) compiled an Adversarial VQA dataset to assess the robustness of VQA models. Schiappa et al. (2022) examined the robustness of video-text models under perturbations, focusing solely on a video-text retrieval task. Furthermore, Qiu et al. (2023d) investigated the robustness of image-text models to perturbations in both modalities across five downstream tasks, while Chen et al. (2023b) evaluated the robustness of adaptation methods across vision-language datasets under multimodal corruptions. More related work can be found in Appendix C.

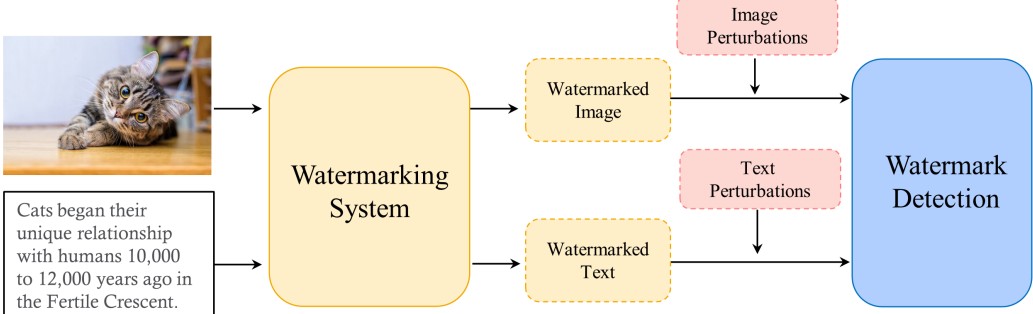

Figure 1: The overall pipeline of our watermarking robustness study. Given a input image/text generated by generative models, the watermarking system adds image/text watermarks to the generated content. Then we conduct watermark detection to evaluate their robustness under image corruptions or text perturbations.

## 3 Preliminary: Invisible Watermark Detection

Before diving into methods for measuring the robustness of watermarks, it is essential to first understand the fundamentals of invisible watermarks and the techniques employed to detect them.

**Definition 1 (Invisible watermark)** *Let $x \in \mathcal{X}$ represent the original content, and let $x_w = \mathsf{Watermark}(x, \mathsf{aux})$ denote the watermarked content, where the watermarking scheme is a function of $x$ and any auxiliary information $\mathsf{aux}$, such as a secret key (Zhao et al., 2023b). A watermark is defined as $\Delta$-invisible on a clean image $x$ with respect to a "distance" function $\mathsf{dist} : \mathcal{X} \times \mathcal{X} \to \mathbb{R}_+$, if $\mathsf{dist}(x, x_w) \leq \Delta$.*

**Definition 2 (Watermark detection)** *A watermark detection algorithm $\mathsf{Detect} : \mathcal{X} \times \mathsf{aux} \to \{0, 1\}$ is designed to determine whether a content $\tilde{x} \in \mathcal{X}$ is watermarked, using auxiliary information such as a secret key ($\mathsf{aux}$) (Zhao et al., 2023b). The algorithm $\mathsf{Detect}$ is subject to two types of errors: false positives, where unwatermarked content is incorrectly classified as watermarked, and false negatives, where watermarked content is incorrectly classified as unwatermarked. We define the content $\tilde{x}$ as being drawn from either the null distribution $P_0$ or the watermarked distribution $P_1$. The Type I error, or false positive rate, is denoted as $\epsilon_1 := \mathrm{Pr}_{x \sim P_0}[\mathsf{Detect}(x) = 1]$, and the Type II error, or false negative rate, is denoted as $\epsilon_2 := \mathrm{Pr}_{x \sim P_1}[\mathsf{Detect}(x) = 0]$.*

A watermarking scheme is typically engineered so that the distribution of watermarked content, $P_1$, is distinct from that of unwatermarked content, $P_0$. This distinction allows the corresponding detection algorithm, which is carefully designed, to almost perfectly differentiate between the two, aiming to ensure that both Type I error ($\epsilon_1$) and Type II error ($\epsilon_2$) are nearly zero. An attack on a watermarking scheme typically involves post-processing a possibly watermarked image in a way that alters both $P_0$ and $P_1$, with the goal of simultaneously increasing both the Type I and Type II errors, thereby evading detection.

## 4 Perturbation Methods

To evaluate the robustness of watermarks under image and text perturbations, we build a comprehensive evaluation benchmark via perturbing the watermarked, generated text or image.

**Image Perturbation.** To simulate real-world corruptions in image data, we employ perturbation strategies adopted from Hendrycks and Dietterich (2019); Qiu et al. (2023d); Zhang et al. (2024). These perturbations are categorized into five groups: **noise, blur, weather, digital**, and **geometric**. Specifically, we utilize 20 different image perturbation techniques across these five categories:

- (1) Noise – Gaussian noise, shot noise, impulse noise, speckle noise;

- (2) Blur – defocus blur, frosted glass blur, motion blur, zoom blur;

- (3) Weather – snow, frost, fog, brightness;

- (4) Digital – contrast, elastic transformation, pixelation, JPEG compression;

- (5) Geometric – scaling, rotation, shearing, piecewise affine transformation.

Acknowledging that real-world corruptions vary in intensity, we introduce variations for each type of corruption as suggested in Hendrycks and Dietterich (2019); Geirhos et al. (2019); Michaelis et al. (2019). In our evaluation, each category features 5 severity levels, culminating in a total of 100 perturbation methods. These strategies, commonly regarded as synthetic distribution shifts, provide a well-defined and manageable starting point. A detailed description of each perturbation method is provided in Table 3 in Appendix A.

**Text Perturbation.** To simulate distribution shifts in language data, we have designed 19 text perturbation techniques organized into three categories: **character-level, word-level**, and **sentence-level** (Qiu et al., 2023d).

- For character-level perturbations, we adopt six strategies from Ma (2019); Qiu et al. (2023d) that simulate common typing errors: keyboard typos, OCR errors, character insertion (CI), character replacement (CR), character swap (CS), and character deletion (CD).

- At the word-level, five strategies from EDA and AEDA (Wei and Zou, 2019; Karimi et al., 2021) are used: synonym replacement (SR), word insertion (WR), word swap (WS), word deletion (WD), and punctuation insertion (IP). These techniques reflect various editorial changes that mimic different writing habits.

- For sentence-level perturbations, eight strategies are included to address more complex linguistic variations: Formal, Casual, Passive, and Active transformations from Li et al. (2018); Etinger and Black (2019); Schmidt (2020); Schiappa et al. (2022) alter the style of the text; Back Translation from Ma (2019); SCPN from Iyyer et al. (2018); BART from Lewis et al. (2019); and DIPPER from (Krishna et al., 2023), focus on semantic shifts due to translation errors and paraphrasing.

Similar to image perturbations, each text perturbation strategy is assigned severity levels. Character-level and word-level perturbations include five severity levels, mirroring the approach used for image perturbations, whereas sentence-level perturbations are applied at a single severity level. In total, this results in 63 text perturbation methods. These techniques encompass a broad range of real-world text distribution shifts—such as typos, word swaps, and style changes Detailed descriptions of each perturbation method are provided in Table 4 in Appendix A.

## 5 Watermarks

In this study, we utilize 4 image watermarking methods and 4 text watermarking methods to evaluate their robustness. The subsequent sections provide a brief introduction to each of these methods.

**Image Watermarks (Image-WM)**

- **DwtDctSvd-WM** Cox et al. (2007) integrates Discrete Wavelet Transform (DWT), Discrete Cosine Transform (DCT), and Singular Value Decomposition (SVD) to embed watermarks in color images. It starts by converting the RGB cover image to YUV, applies DWT to the Y channel, segments it into blocks via DCT, and performs SVD on each block before embedding the watermark. DwtDctSvd is the default method used by Stable Diffusion (Rombach et al., 2022).

- **RivaGAN-WM** Zhang et al. (2019) introduces a robust image watermarking technique utilizing Generative Adversarial Networks (GANs) (Goodfellow et al., 2014). It incorporates two adversarial networks: one to evaluate the quality of watermarked images and another to facilitate watermark removal. The system includes an encoder for watermark embedding and a decoder for its extraction, enhancing both performance and robustness. RivaGAN is also employed as a watermarking method by Stable Diffusion (Rombach et al., 2022).

- **SSL-WM** Fernandez et al. (2022b) leverages the latent spaces of pre-trained neural networks for watermark encoding, using networks trained via self-supervised learning (SSL) to capture effective watermarking features. This method embeds watermarks by applying backpropagation and data augmentation, and it is capable of detecting and decoding these watermarks from the watermarked image or its extracted features.

- **StegaStamp-WM** Tancik et al. (2019) introduces a learned steganographic algorithm designed for the robust encoding and decoding of arbitrary hyperlink bit strings into photos, achieving near-perceptual invisibility. The system utilizes a deep neural network to learn an encoding/decoding algorithm that remains robust against image perturbations typical of real-world printing and photography scenarios.

**Text Watermarks (Text-WM)**

- **KGW-WM** Kirchenbauer et al. (2023a) randomly splits the vocabulary into red tokens and green tokens based on the hash value of the previous token. During the next token generation, a constant $\delta$ is added to the logits for tokens that belong to the green

list. This effectively increases the probability of generating green list tokens, thereby increasing the overall number of green tokens in the entire output sequence. During detection, if the suspect text contains significantly more green tokens, it is likely from the watermarked LLM.

- **KTH-WM** obtains a watermark key from the watermark sequence and generates the watermarked text by mapping the watermark key (random numbers) to the sample from the language model. Kuditipudi et al. (2023) provided two instantiations of this protocol, namely inverse transform sampling and exponential minimum sampling. Both schemes ensure distortion-free – the expected distribution of a single response from the watermarked model is identical to the distribution of a single response from the original model. We utilize exponential minimum sampling for our experiments due to its stronger reported results (Kuditipudi et al., 2023).

- **Blackbox-WM** Yang et al. (2023): begins by obtaining the original text from a black-box language model. The process involves selectively replacing words with context-based synonyms to embed a watermark. It employs a unique binary encoding function that assigns a random binary value (either bit-0 or bit-1) to each word. This function ensures a near balance between bit-0 and bit-1 representations in non-watermarked texts. For every word selected for replacement, the method generates synonym candidates, each evaluated for the binary encoding they carry.

- **Unigram-WM** Zhao et al. (2023a) involves a watermarking process similar to KGW-WM, splitting the vocabulary into the green list and the red list and then increasing the probability of generating green tokens. The key difference is that Unigram-WM keeps the red-green partitions fixed. This allows for better robustness guarantees since each edit (insertion/deletion/replacement) only changes one token from green to red or from red to green.

## 6 Experiments

In this study, we aim to address several key questions: (1) Which watermarking methods demonstrate the highest stability and robustness? (2) Which perturbation techniques are most effective in attacking the embedded watermarks? (3) Among image and text generation models, which are the most robust?

### 6.1 Experimental Settings

**Benchmark Models** We investigate the following benchmark models, all of which are publicly available. Each model has been chosen for its relevance and potential to provide insights into the watermark robustness against perturbations. We used 16 NVIDIA A6000 GPUs for our experiments.
- Text-to-image models: NExT-GPT (Wu et al., 2023), Stable Diffusion (Rombach et al., 2021), DALLE3 (Betker et al.), SDXL-Lightning (Lin et al., 2024), PIXART (Chen et al., 2023a), Kandinsky 2.2 (Razzhigaev et al., 2023), Latent Consistency Models (LCMs) (Luo et al., 2023), RPG (Yang et al., 2024).

- Image-to-text models: NExT-GPT (Wu et al., 2023), Fuyu-8B (Bavishi et al., 2023), InternLM-XComposer (Zhang et al., 2023), InstructBLIP (Dai et al., 2023), LLaVA 1.6 (Liu et al., 2023), MiniGPT-4 (Zhu et al., 2023), mPLUG-Owl2 (Ye et al., 2023), Qwen-VL (Bai et al., 2023).

**Evaluation Metrics**   For evaluating image quality, our study employs several metrics: Peak Signal-to-Noise Ratio (PSNR), Structural Similarity Index Measure (SSIM) (Wang et al., 2004), Bit Accuracy (Bit Acc), and Detection Accuracy (Dect Acc) (Zhao et al., 2023a). In assessing text quality, we utilize BLEURT (Sellam et al., 2020; Pu et al., 2021), ROUGE (Lin, 2004), Bit Accuracy (Bit Acc), and Detection Accuracy (Dect Acc) (Zhao et al., 2023a). These metrics allow us to assess the fidelity and integrity of watermarked images comprehensively.

## 6.2 Results And Discussions

Table 1: Comparison of different image watermarking methods.

| Watermark | Model | PSNR | SSIM | Bit Acc | Dect Acc | Dect Acc (Ori) | Dect Acc Drop (%) |
|---|---|---|---|---|---|---|---|
| DctDwtSvd-WM | NextGPT | 17.03 | 0.50 | 4.29 | 12.12 | 95.56 | -87.32% |
| | Stable Diffusion | 17.69 | 0.51 | 4.43 | 15.79 | 100.00 | -84.21% |
| | DALLE3 | 16.85 | 0.51 | 3.09 | 15.81 | 95.32 | -83.41% |
| | SDXL-Lightning | 18.55 | 0.56 | 5.96 | 21.27 | 100.00 | -78.73% |
| | PIXART | 17.66 | 0.54 | 4.28 | 18.83 | 95.57 | -80.30% |
| | Kandinsky 2.2 | 15.99 | 0.50 | 4.01 | 7.51 | 91.42 | -91.79% |
| | LCMs | 19.48 | 0.58 | 6.32 | 18.81 | 100.00 | -81.19% |
| | RPG-Image | 17.74 | 0.50 | 3.79 | 29.19 | 92.20 | -68.34% |
| RivaGAN-WM | NextGPT | 17.60 | 0.54 | 3.87 | 23.40 | 98.55 | -76.26% |
| | Stable Diffusion | 17.19 | 0.51 | 3.10 | 31.39 | 100.00 | -68.61% |
| | DALLE3 | 16.93 | 0.52 | 3.59 | 25.40 | 97.59 | -73.97% |
| | SDXL-Lightning | 18.40 | 0.55 | 3.56 | 36.83 | 100.00 | -63.17% |
| | PIXART | 17.82 | 0.55 | 7.01 | 33.10 | 96.63 | -65.75% |
| | Kandinsky 2.2 | 16.05 | 0.51 | 5.53 | 17.17 | 90.33 | -80.99% |
| | LCMs | 19.28 | 0.57 | 3.83 | 35.84 | 99.00 | -63.80% |
| | RPG-Image | 17.30 | 0.50 | 4.29 | 29.55 | 91.63 | -67.75% |
| SSL-WM | NextGPT | 7.71 | 0.30 | 7.05 | 26.18 | 96.67 | -72.92% |
| | Stable Diffusion | 8.08 | 0.31 | 2.51 | 24.72 | 95.42 | -74.09% |
| | DALLE3 | 9.91 | 0.27 | 3.62 | 25.67 | 88.54 | -71.01% |
| | SDXL-Lightning | 10.65 | 0.32 | 3.62 | 28.14 | 86.48 | -67.46% |
| | PIXART | 8.54 | 0.30 | 3.09 | 24.86 | 92.41 | -73.10% |
| | Kandinsky 2.2 | 8.46 | 0.24 | 3.90 | 20.91 | 90.85 | -76.98% |
| | LCMs | 8.75 | 0.34 | 3.06 | 12.01 | 94.87 | -87.34% |
| | RPG-Image | 10.54 | 0.30 | 3.40 | 23.70 | 90.18 | -73.72% |
| StegaStamp-WM | NextGPT | 7.33 | 0.24 | 3.49 | 14.12 | 85.73 | -83.53% |
| | Stable Diffusion | 7.38 | 0.29 | 4.04 | 19.26 | 82.29 | -76.59% |
| | DALLE3 | 6.83 | 0.29 | 3.84 | 15.98 | 88.41 | -81.93% |
| | SDXL-Lightning | 6.6 | 0.22 | 4.76 | 9.25 | 84.72 | -89.08% |
| | PIXART | 8.15 | 0.21 | 3.31 | 14.53 | 85.69 | -83.04% |
| | Kandinsky 2.2 | 6.91 | 0.22 | 3.68 | 9.64 | 86.35 | -88.84% |
| | LCMs | 6.94 | 0.21 | 4.15 | 25.71 | 88.67 | -85.78% |
| | RPG-Image | 5.81 | 0.26 | 3.46 | 8.74 | 80.26 | -89.11% |

**Watermarking Strategy Comparison.**   In Tables 1 and 2, we present the outcomes of various image and text watermarking methods, respectively. The results indicate variations in the performance across different models. Generally, under image perturbations, RivaGAN-

Table 2: Comparison of different text watermarking methods.

| Watermark | Model | BLEURT | ROUGE | Bit Acc | Dect Acc | Dect Acc (ori) | Dect Acc Drop (%) |
|---|---|---|---|---|---|---|---|
| KGW-WM | NextGPT | 0.32 | 39.54 | 59.41 | 19.60 | 99.76 | -80.35% |
| | Fuyu | 0.30 | 40.39 | 67.87 | 36.27 | 99.85 | -63.68% |
| | InternLM-XComposer | 0.31 | 35.07 | 64.68 | 14.97 | 100.00 | -85.03% |
| | InstructBLIP | 0.30 | 3.07 | 51.26 | 23.60 | 100.00 | -76.40% |
| | LLaVA 1.5 | 0.35 | 42.34 | 68.11 | 71.38 | 98.04 | -37.93% |
| | MiniGPT-4 | 0.37 | 43.21 | 57.84 | 18.95 | 99.48 | -80.95% |
| | mPLUG-Owl2 | 0.30 | 43.17 | 59.78 | 37.64 | 99.95 | -62.34% |
| | Qwen-VL | 0.29 | 29.16 | 63.09 | 23.10 | 99.93 | -76.88% |
| KTH-WM | NextGPT | 0.31 | 41.43 | 51.78 | 21.06 | 98.37 | -78.59% |
| | Fuyu | 0.29 | 41.50 | 66.34 | 33.50 | 99.06 | -66.18% |
| | InternLM-XComposer | 0.23 | 38.13 | 57.86 | 33.00 | 100.00 | -67.00% |
| | InstructBLIP | 0.36 | 14.54 | 56.34 | 34.00 | 99.85 | -65.95% |
| | LLaVA 1.5 | 0.31 | 31.74 | 64.90 | 35.75 | 98.68 | -63.77% |
| | MiniGPT-4 | 0.32 | 44.14 | 50.63 | 21.50 | 100.00 | -78.50% |
| | mPLUG-Owl2 | 0.27 | 36.63 | 62.55 | 33.41 | 99.64 | -66.47% |
| | Qwen-VL | 0.25 | 35.52 | 54.23 | 32.59 | 99.87 | -67.37% |
| Blackbox-WM | NextGPT | 0.33 | 43.09 | 62.41 | 23.80 | 100.00 | -76.20% |
| | Fuyu | 0.30 | 40.01 | 66.12 | 27.93 | 100.00 | -72.07% |
| | InternLM-XComposer | 0.28 | 33.44 | 52.75 | 25.40 | 99.57 | -74.49% |
| | InstructBLIP | 0.37 | 35.51 | 53.24 | 22.70 | 99.86 | -77.27% |
| | LLaVA 1.5 | 0.31 | 42.22 | 60.60 | 22.72 | 99.62 | -77.19% |
| | MiniGPT-4 | 0.37 | 44.72 | 64.53 | 24.01 | 100.00 | -75.99% |
| | mPLUG-Owl2 | 0.30 | 34.81 | 62.54 | 31.58 | 99.86 | -68.38% |
| | Qwen-VL | 0.28 | 33.19 | 51.46 | 24.68 | 99.68 | -75.24% |
| Unigram-WM | NextGPT | 0.26 | 26.39 | 48.42 | 2.46 | 100.00 | -97.54% |
| | Fuyu | 0.24 | 28.14 | 48.98 | 0.77 | 100.00 | -99.23% |
| | InternLM-XComposer | 0.19 | 25.37 | 47.47 | 0.32 | 100.00 | -99.68% |
| | InstructBLIP | 0.25 | 17.18 | 47.88 | 17.74 | 99.76 | -82.22% |
| | LLaVA 1.5 | 0.37 | 40.01 | 58.16 | 4.68 | 99.69 | -95.31% |
| | MiniGPT-4 | 0.32 | 32.67 | 45.33 | 2.48 | 98.62 | -97.49% |
| | mPLUG-Owl2 | 0.23 | 28.08 | 47.58 | 1.27 | 97.42 | -98.70% |
| | Qwen-VL | 0.21 | 24.99 | 42.57 | 0.28 | 99.03 | -99.72% |

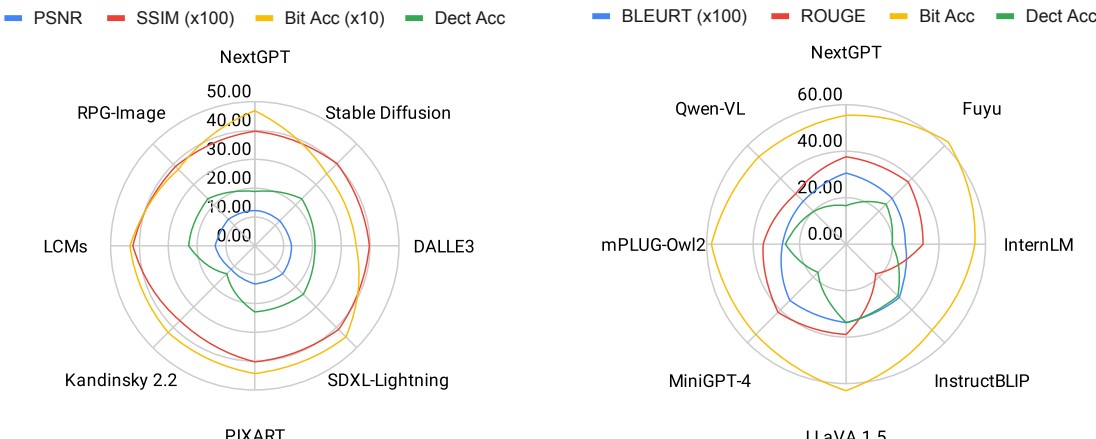

Figure 2: Performance comparison of different models under image perturbations.

Figure 3: Performance comparison of different models under text perturbations.

WM appears to be more robust, whereas under text perturbations, KGW-WM demonstrates greater stability.

**Perturbation Method Comparison.** In Figure 4, we showed the comparisons of different [Top] image corruption and [Bottom] text perturbation methods. All the results have been averaged on different severity levels.

For image corruptions, we show the results by Stable Diffusion in Figure 4 [Top]. Generally, noise and blur-based perturbations tend to have a more severe impact on all metrics, as they directly affect the clarity and sharpness of images. Environmental effects and quality degradation also impact the metrics but might be less severe compared to direct noise introductions or blurs. Gaussian, Shot, Impulse, and Speckle noise appear to significantly decrease PSNR and SSIM, indicating a substantial degradation in image quality. JPEG compression, while degrading quality, may not affect structural similarity as much, depending on the compression level. Distortions such as Rotate, Shear, and Piecewise Affine could particularly lower Detection Accuracy as they alter the geometry and spatial relationships within the image, potentially complicating detection tasks. Based on the results in Figure 4 [Top], we find that Zoom Blur is more effective, and Glass Blur is less effective.

Similarly, for text perturbations, we show the results by LLaVA 1.6 in Figure 4 [Bottom]. Stylistic transformations (Formal, Casual, Active, Passive) show the highest performance across all metrics, indicating minimal impact on text quality and excellent preservation and detection capabilities. Synonym Replacement (SR), Bart, SPCN, and Dipper also perform well, with high BLEURT and ROUGE scores and good preservation and detection accuracy. Character Insert (CI), Character Delete (CD), and Word Swap (WS) show the lowest scores across all metrics, indicating significant degradation in text quality and moderate preservation and detection capabilities. Based on the results in Figure 4 [Bottom], we find that character-level perturbations are more effective, and sentence-level perturbations are less effective.

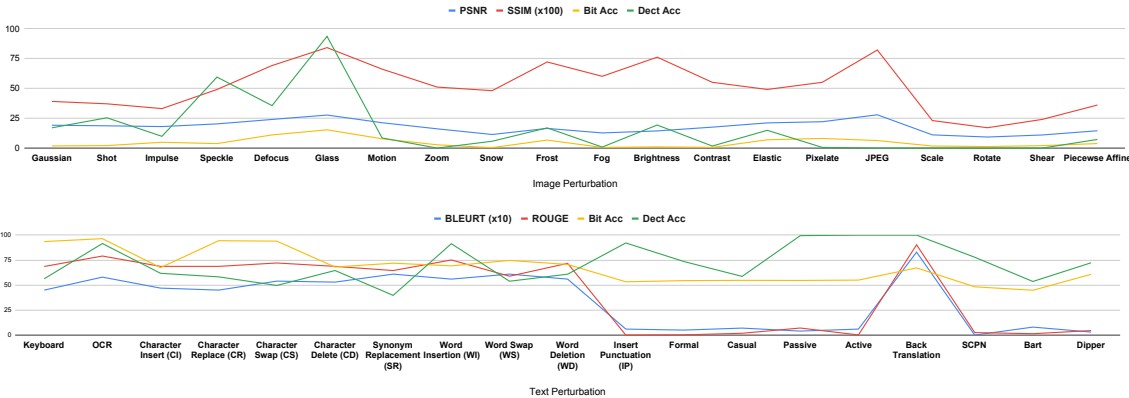

Figure 4: Comparisons of different [Top] image corruption and [Bottom] text perturbation methods using Stable Diffusion and LLaVA, respectively. All the results were averaged at different severity levels.

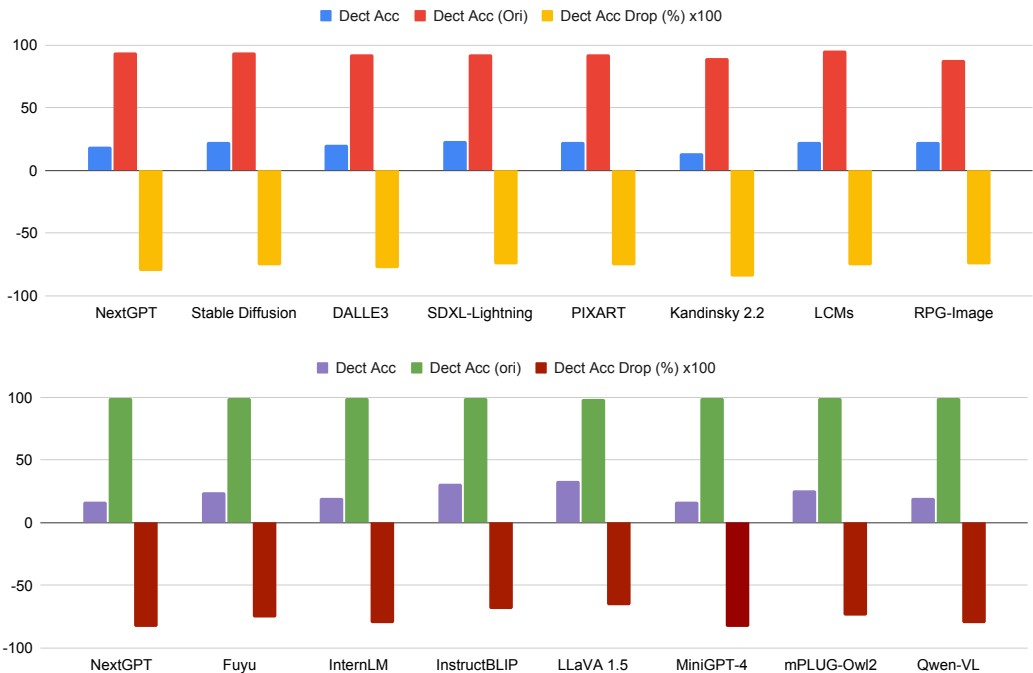

Figure 5: Model comparisons under [Top] image corrections and [Bottom] text perturbations. All the results have been averaged on the performance under all image/text perturbations.

**Model Comparison.** In Figure 5, we showed the model comparisons under [Top] image and [Bottom] text perturbations. The results are averaged across all image and text perturbations. Our findings indicate that under image perturbations, SDXL-Lightning demonstrates superior robustness. Conversely, LLaVA exhibits greater robustness under text perturbations compared to the other models.

### 6.3 Ablation Study

**Image Perturbation Severity Influence.** Each image perturbation method is associated with multiple severity levels, so we would like to explore the relationship between robustness performance and perturbation severity levels. In Figure 6, we showed two examples, Gaussian Noise and Glass Blur, across varying levels of severity (from 1 to 5). PSNR and SSIM are particularly sensitive to distortions, with PSNR showing a more pronounced drop in the presence of Gaussian Noise than Glass Blur. This suggests that noise introduces more disruptive interference compared to blur. SSIM, while also decreasing with severity, indicates that structural elements of images are somewhat more preserved under Glass Blur compared to Gaussian Noise, highlighting differential impacts depending on the type of distortion. Bit Accuracy exhibits a notable decline under both conditions but is more affected by Glass Blur, especially beyond moderate levels of severity. This suggests that blur more significantly affects the bit-level representation of the image. Detection accuracy remains relatively stable under Glass Blur and shows resilience, suggesting that detection algorithms might still

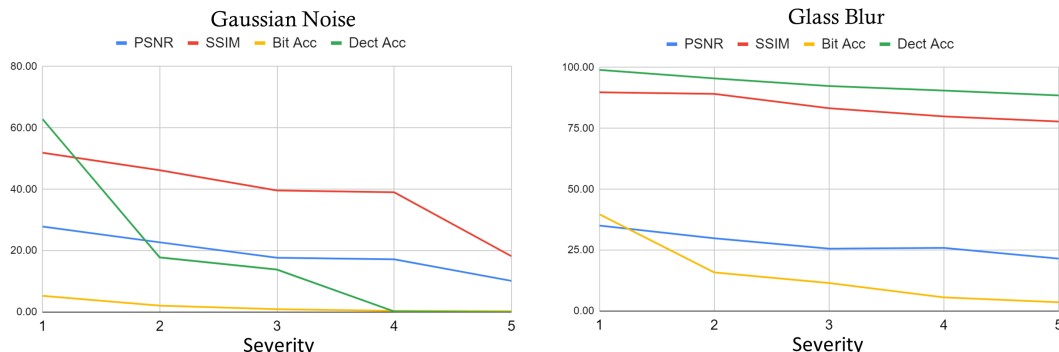

Figure 6: Performance changes with different severity levels under image perturbations.

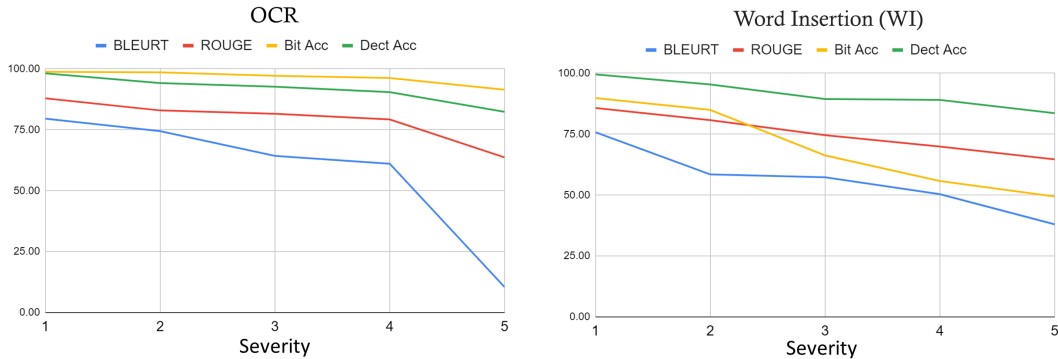

Figure 7: Performance changes with different severity levels under text perturbations.

effectively identify key features in blurred images. However, it declines under Gaussian Noise, indicating challenges in feature detection amidst this type of noise.

**Text Perturbation Severity Influence.** Similar to the image perturbations above, in Figure 7, we showed two examples, OCR and Word Insertion, across varying levels of severity (from 1 to 5). OCR errors cause a more drastic decrease in BLEURT scores than word insertions, suggesting that OCR errors might lead to more severe semantic disruptions than simple word additions. Detection accuracy demonstrates notable resilience in both scenarios, implying that the underlying algorithms are effective at extracting essential information despite textual distortions.

### 6.4 Limitations and Future Work.

- Our current evaluation study focuses on watermark robustness against image corruptions and text perturbations. However, it does not encompass other watermark types, such as audio, nor does it include all potential perturbations. For instance, while there are studies on the robustness against adversarial image perturbations, such attacks typically involve a classification task conditioned on a target label, making them inapplicable to our evaluation setting. Therefore, they were not included in this study.

- Our study investigated 8 image-to-text models and 8 text-to-image models. However, we acknowledge that this is a preliminary study. Different watermarking strategies may show

varying performance when the embedded watermarks change. Consequently, the findings related to the models are speculative and not conclusive.

- We hope that future research will expand the evaluation benchmark to include a broader range of watermark types and perturbations. To the best of our knowledge, our study does not pose any potential negative societal impacts.

> Takeaway: Our main findings are as follows.
> (1) Watermarks are sensitive to image corruptions and text perturbations.
> (2) Among image perturbations, Zoom Blur consistently shows the highest impact, while Glass Blur is the least harmful.
> (3) Among text perturbations, character-level perturbations are more effective, and sentence-level perturbations are less effective.
> (4) Under image perturbations, RivaGAN-WM appears more stable, whereas under text perturbations, KGW-WM seems more stable.
> (5) In terms of models, SDXL-Lightning is more robust than other baselines under image perturbations, while LLaVA demonstrates greater robustness under text perturbations.

## 7 Conclusion

In this study, we explore the robustness of image and text watermarks under perturbations. Our research includes testing the performance of 8 image-to-text models and 8 text-to-image models, subjected to 100 image perturbation techniques and 63 text perturbation methods. We assess the robustness of 4 image watermarking methods and 4 text watermarking methods. We aim for our benchmark to be a valuable resource for examining the robustness of image and text watermarks, and we hope our results will inspire the development and implementation of more robust watermarking strategies for practical applications.

## Broader Impact Statement

**Positive Impacts:**

- Copyright Protection: By improving watermarking techniques, our research helps in safeguarding the intellectual property rights of creators, particularly in digital media. This can lead to more secure ways for artists, writers, and developers to claim ownership and receive proper attribution.

- Content Authenticity: As fake content proliferates, robust watermarking can be crucial in verifying the authenticity of digital media. This could be particularly significant for news agencies, educational content providers, and other stakeholders interested in preserving the integrity of information.

- Content Management: Effective watermarking aids content providers in tracking and managing the distribution of their work. This can help in enforcing licensing agreements and preventing unauthorized use, thus potentially reducing copyright violations.

**Negative Impacts:**

- Privacy Concerns: Robust watermarking could potentially be used for surveillance and tracking purposes. For example, watermarked images or texts could be used to trace the activity of individuals without their consent.

- Accessibility and Misuse: As watermarking techniques become more sophisticated, there might be a risk of these technologies being misused to embed malicious data or to restrict access to content via overly aggressive copyright enforcement.

- Economic Impact: There may be concerns regarding the economic implications for businesses that rely on less secure watermarking methods, which might face obsolescence or the need for costly upgrades.

We acknowledge the dual-edged nature of technological advancements in watermarking, emphasizing a commitment to ethical considerations and the responsible deployment of these technologies to maximize benefits while minimizing harm.

**Ethical Discussion:** In addition to the positive and negative impacts discussions, we also discuss the ethical implications of our research as follows:

- Equity and Accessibility: Our techniques could inadvertently favor large organizations by enhancing their ability to protect intellectual property, potentially marginalizing smaller entities and researchers. This could concentrate power within large tech companies, thus impacting the balance between protection and accessibility in AI development.

- Potential for Misuse: While watermarking can aid in identifying sources of malicious content and facilitate moderation, it also poses risks if exploited by bad actors to claim false ownership or bypass protections. Furthermore, robust watermarking could be misused for surveillance, raising significant privacy and ethical concerns.

- Trade-offs: Implementing these watermarking systems involves balancing protection strength, model performance, and accessibility. It is crucial to consider these trade-offs carefully to avoid stifling innovation or restricting access to beneficial AI technologies.

## Acknowledgments and Disclosure of Funding

This study is supported in part by Google Research, Carnegie Mellon University Computer Science Department fellowship, and a CMU CyLab Seed Grant. We would like to thank the feedback from Sven Gowal, Yonatan Bisk, Daniel Fried, and William Wang.

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

## Appendix A. More Details about Perturbation Methods

Table 3 shows a detailed introduction to each image perturbation method. Table 4 shows a detailed introduction about each text perturbation method.

## Appendix B. Experimental Settings

### B.1 Image Watermark Parameters

**DctDwtSvd-WM and RivaGAN-WM**  For all of our experiments, we embed 4 characters as bytes, specifically the string "test", into the images. During decoding, we deem the watermark detection algorithm as passing if it is able to fully retrieve the original, encoded string "test".

**SSL-WM**  A neural network is needed to extract the features from images, and a normalization layer is to evenly distribute the extracted features in the latent space. We utilize the recommended, default model and normalization layers, namely ResNet-50 trained with DINO and PCA whitening respectively. During detection, we consider the zero-bit scenario (Fernandez et al., 2022b).

**StegaStamp-WM**  requires a pretrained encoder, decoder, and detector. The encoder is an architecture similar to U-Net (Ronneberger et al., 2015) where the image is first processed through a fully connected network to become a tensor of size $50 \times 50 \times 3$, then upsampled to get a tensor of size $400 \times 400 \times 3$. The decoder is a spatial transformer network (Jaderberg et al., 2015) that is trained to recover the encoded watermark. For the detector, the authors use an open-source semantic segmentation network, namely BiSeNet (Yu et al., 2018). We utilize all of the default parameters used in (Tancik et al., 2019).

### B.2 Text Watermark Parameters

**KGW-WM**  We utilize game and delta values of 0.25 and 2.0, respectively. We also set the seeding scheme as 'simple_1', which represents a simple bigram hash, to utilize the main settings of the experiments in the paper (Kirchenbauer et al., 2023a). During detection, we also utilize a $z$ threshold of 0.5 and ignore repeated n-grams.

**KTH-WM**  We set the desired length of the generated text, $m$, to 30 as detailed in Kuditipudi et al. (2023). The length of the watermark sequence, $n$, is kept at the standard value of 256. For generating the random watermark sequence, we employ a key of 42. The authors' method of evaluating their watermarking framework involves p-values, and we consider texts with $p < 0.1$ as effectively watermarked.

**Blackbox-WM**  We employ the $\tau$ word value of 0.8, and a $\lambda$ value of 0.83 (Yang et al., 2023). We also use the "embed" mode during watermarking. During detection, if the confidence value is over 80%, we deem the detection algorithm as successfully finding the watermark.

**Unigram-WM**  When applying the watermark, we utilize a fraction and strength value of 0.5 and 2.0, respectively. Additionally, we determined the watermark key to be defaulted to 0. During detection, we utilize the default value of 6.0 (Zhao et al., 2023a).

Table 3: Image perturbations.

| Category | Perturbation | Description | Severities |
|---|---|---|---|
| Noise | Gaussian Noise | Gaussian noise can appear in low-lighting conditions. | 5 |
| | Shot Noise | Shot noise, also called Poisson noise, is electronic noise caused by the discrete nature of light itself. | 5 |
| | Impulse Noise | Impulse noise is a color analogue of salt-and-pepper noise and can be caused by bit errors. | 5 |
| | Speckle Noise | Speckle noise is the noise added to a pixel that tends to be larger if the original pixel intensity is larger. | 5 |
| Blur | Defocus Blur | Defocus blur occurs when an image is out of focus. | 5 |
| | Frosted Glass Blur | Frosted Glass Blur appears with "frosted glass" windows or panels. | 5 |
| | Motion Blur | Motion blur appears when a camera is moving quickly. | 5 |
| | Zoom Blur | Zoom blur occurs when a camera moves toward an object rapidly. | 5 |
| Weather | Snow | Snow is a visually obstructive form of precipitation. | 5 |
| | Frost | Frost forms when lenses or windows are coated with ice crystals. | 5 |
| | Fog | Fog shrouds objects and is rendered with the diamond-square algorithm. | 5 |
| | Brightness | Brightness varies with daylight intensity. | 5 |
| Digital | Contrast | Contrast can be high or low depending on lighting conditions and the photographed object's color. | 5 |
| | Elastic | Elastic transformations stretch or contract small image regions. | 5 |
| | Pixelate | Pixelation occurs when upsampling a low-resolution image. | 5 |
| | JPEG Compression | JPEG is a lossy image compression format that introduces compression artifacts. | 5 |
| Geometric | Scale | Change the size of an image by enlarging or shrinking its dimensions. | 5 |
| | Rotate | Turn the image around a central point by a specified degree, altering its orientation. | 5 |
| | Shear | Skew the image by shifting parts of it more than others, creating a distortion. | 5 |
| | Piecewise Affine | Apply affine transformations to different parts of the image independently, allowing for complex local distortions. | 5 |
| Sum | **20** | — | **100** |

Table 4: Text perturbations.

| Category | Perturbation | Description | Severities |
|---|---|---|---|
| Character-level | Keyboard | Substitute character by keyboard distance with probability $p$. | 5 |
| | OCR | Substitute character by pre-defined OCR error with probability $p$. | 5 |
| | Character Insert (CI) | Insert character randomly with probability $p$. | 5 |
| | Character Replace (CR) | Substitute character randomly with probability $p$. | 5 |
| | Character Swap (CS) | Swap character randomly with probability $p$. | 5 |
| | Character Delete (CD) | Delete character randomly with probability $p$. | 5 |
| Word-level | Synonym Replacement (SR) | Randomly choose $n$ words from the sentence that are not stop words. Replace each of these words with one of its synonyms chosen at random. | 5 |
| | Word Insertion (WI) | Find a random synonym of a random word in the sentence that is not a stop word. Insert that synonym into a random position in the sentence. Do this $n$ times. | 5 |
| | Word Swap (WS) | Randomly choose two words in the sentence and swap their positions. Do this $n$ times. | 5 |
| | Word Deletion (WD) | Each word in the sentence can be randomly removed with probability $p$. | 5 |
| | Insert Punctuation (IP) | Random insert punctuation in the sentence with probability $p$. | 5 |
| Sentence-level | Formal | Transfer the text style to Formal. | 1 |
| | Casual | Transfer the text style to Casual. | 1 |
| | Passive | Transfer the text style to Passive. | 1 |
| | Active | Transfer the text style to Active. | 1 |
| | Back Translation (BT) | Translate source to German and translate it back to English via (Ng et al., 2020). | 1 |
| | SCPN | Produce a paraphrase of a given sentence with specified syntactic structures (Iyyer et al., 2018). | 1 |
| | BART | Use BART for text summarization as paraphrasing attack (Lewis et al., 2019). | 1 |
| | DIPPER | DIPPER can paraphrase paragraphs, condition on surrounding context, and control lexical diversity and content reordering (Krishna et al., 2023). | 1 |
| Sum | **19** | — | **63** |

## B.3 Number of Samples

We utilized 5,000 image-caption pairs from the COCO validation split (Lin et al., 2014). For text generation, the input to the multimodal models consisted of the prompt "Describe this image:" alongside the corresponding image from the dataset. For image generation, the input was the prompt "Please generate an image describing the following caption: `C`", where `C` is the corresponding caption from the dataset. In total, we generated 5,000 images and 5,000 texts for each model.

## Appendix C. More Related Work

**Multimodal Watermarking**   is a technique that embeds watermarks into various types of media content like audio, video, and images. (Tang et al., 2023) proposed a safe and robust backdoor-based embedding watermarking method for VLPs called VLPMarker. It further proposed a collaborative copyright verification strategy based on both backdoor triggers and embedding distribution, enhancing resilience against various attacks.

**Multimodal Learning**   The study of multimodal learning traces back to 1989, when Yuhas et al. (1989) leveraged the McGurk Effect to explore audio-visual speech recognition using neural networks (Tiippana, 2014; McGurk and MacDonald, 1976). Since then, collaboration between researchers in natural language processing (NLP) and computer vision (CV) has led to the creation of large multimodal datasets designed for tasks such as classification, translation, and detection. Advances in large language models (LLMs) have further facilitated the integration of additional modalities, most notably visual data (Wang et al., 2022; Qiu et al., 2023d; Nguyen et al., 2022; Li et al., 2022; Qiu et al., 2024a; Wang et al., 2021; Qiu et al., 2023c; Shah et al., 2022; Zhang et al., 2021; Qiu et al., 2023a; Xu et al., 2023; Qiu et al., 2024b; Wang et al., 2023; Qiu et al., 2023b; Long and Yao, 2020). By leveraging embeddings pretrained on both language and image datasets, vision-language models achieve exceptional performance across various tasks.

## Appendix D. More Experimental Results

**Trade-off between detection accuracy and content quality**   Figure 6 and Figure 7 illustrate the trade-offs between various metrics such as PSNR, SSIM, BLEURT, ROUGE, Bit Accuracy, and Detection Accuracy, with respect to severity levels of image and text perturbations, respectively. Detailed analyses (omitted from the paper due to their large volume) indicate that SSL-WM and Blackbox-WM provide the optimal balance for image and text watermarking, respectively.

## Appendix E. Detailed Experimental Results

In the following tables, we provide the detailed results for each baseline model mentioned in Section 6.1.

Table 5: NExT-GPT image watermarks under image perturbations.

| Watermark | Attack Category | Perturbation | PSNR | SSIM | Bit Acc | Dect Acc |
|---|---|---|---|---|---|---|
| DctDwtSvd-WM | Noise | Gaussian | 18.95 | 0.40 | 1.87 | 5.68 |
| | | Shot | 18.32 | 0.41 | 2.42 | 9.25 |
| | | Impulse | 18.18 | 0.38 | 4.79 | 1.38 |
| | | Speckle | 0.00 | 0.50 | 3.59 | 25.48 |
| | Blur | Defocus | 22.93 | 0.55 | 7.40 | 28.82 |
| | | Glass | 27.05 | 0.83 | 10.56 | 80.69 |
| | | Motion | 20.53 | 0.61 | 5.92 | 6.75 |
| | | Zoom | 17.31 | 0.50 | 3.63 | 0.00 |
| | Weather | Snow | 11.13 | 0.52 | 0.29 | 6.42 |
| | | Frost | 10.00 | 0.60 | 0.33 | 2.98 |
| | | Fog | 13.22 | 0.59 | 0.70 | 0.38 |
| | | Brightness | 14.03 | 0.77 | 0.88 | 18.75 |
| | Digital | Contrast | 14.73 | 0.53 | 0.88 | 1.48 |
| | | Elastic | 16.70 | 0.41 | 5.02 | 11.49 |
| | | Pixelate | 22.76 | 0.65 | 7.97 | 36.31 |
| | | JPEG | 28.09 | 0.84 | 5.93 | 0.00 |
| | Geometric | Scale | 4.88 | 0.22 | 8.94 | 0.00 |
| | | Rotate | 27.32 | 0.10 | 7.30 | 0.00 |
| | | Shear | 24.54 | 0.27 | 3.31 | 2.31 |
| | | Piecewse Affine | 10.02 | 0.25 | 4.14 | 4.28 |

| Watermark | Attack Category | Perturbation | PSNR | SSIM | Bit Acc | Dect Acc |
|---|---|---|---|---|---|---|
| RivaGAN-WM | Noise | Gaussian | 18.96 | 0.40 | 1.87 | 12.89 |
| | | Shot | 18.31 | 0.41 | 2.36 | 12.08 |
| | | Impulse | 18.18 | 0.38 | 4.78 | 8.20 |
| | | Speckle | 19.88 | 0.51 | 3.58 | 20.71 |
| | Blur | Defocus | 22.94 | 0.55 | 6.87 | 32.69 |
| | | Glass | 27.05 | 0.83 | 9.74 | 84.54 |
| | | Motion | 20.51 | 0.61 | 5.53 | 32.79 |
| | | Zoom | 17.32 | 0.50 | 3.56 | 2.70 |
| | Weather | Snow | 11.17 | 0.52 | 0.29 | 4.38 |
| | | Frost | 9.98 | 0.60 | 0.33 | 12.97 |
| | | Fog | 13.25 | 0.59 | 0.67 | 2.37 |
| | | Brightness | 14.14 | 0.77 | 0.85 | 51.03 |
| | Digital | Contrast | 14.74 | 0.53 | 0.88 | 5.07 |
| | | Elastic | 16.70 | 0.41 | 4.55 | 59.62 |
| | | Pixelate | 22.77 | 0.65 | 7.18 | 42.50 |
| | | JPEG | 28.07 | 0.84 | 5.90 | 0.15 |
| | Geometric | Scale | 15.64 | 0.42 | 5.82 | 1.18 |
| | | Rotate | 14.34 | 0.23 | 6.20 | 72.50 |
| | | Shear | 15.93 | 0.68 | 4.31 | 5.28 |
| | | Piecewse Affine | 12.21 | 0.31 | 2.22 | 4.41 |

| Watermark | Attack Category | Perturbation | PSNR | SSIM | Bit Acc | Dect Acc |
|---|---|---|---|---|---|---|
| SSL-WM | Noise | Gaussian | 8.67 | 0.17 | 1.27 | 2.50 |
| | | Shot | 7.72 | 0.16 | 1.49 | 2.43 |
| | | Impulse | 7.59 | 0.15 | 43.87 | 0.00 |
| | | Speckle | 8.56 | 0.22 | 2.58 | 13.89 |
| | Blur | Defocus | 11.48 | 0.31 | 4.22 | 29.46 |
| | | Glass | 13.62 | 0.42 | 6.55 | 74.71 |
| | | Motion | 10.25 | 0.31 | 3.47 | 22.36 |
| | | Zoom | 8.64 | 0.25 | 2.08 | 0.04 |
| | Weather | Snow | 5.56 | 0.26 | 0.21 | 3.23 |
| | | Frost | 4.97 | 0.30 | 0.20 | 29.71 |
| | | Fog | 6.58 | 0.30 | 0.32 | 59.22 |
| | | Brightness | 7.04 | 0.39 | 1.12 | 73.01 |
| | Digital | Contrast | 7.36 | 0.27 | 0.49 | 55.45 |
| | | Elastic | 8.34 | 0.21 | 2.88 | 46.66 |
| | | Pixelate | 11.38 | 0.33 | 4.67 | 5.60 |
| | | JPEG | 14.00 | 0.42 | 3.19 | 2.12 |
| | Geometric | Scale | 3.77 | 0.48 | 26.91 | 25.67 |
| | | Rotate | 2.91 | 0.25 | 2.93 | 30.82 |
| | | Shear | 2.63 | 0.35 | 4.01 | 24.19 |
| | | Piecewse Affine | 3.14 | 0.41 | 28.53 | 22.58 |

| Watermark | Attack Category | Perturbation | PSNR | SSIM | Bit Acc | Dect Acc |
|---|---|---|---|---|---|---|
| StegaStamp-WM | Noise | Gaussian | 8.41 | 0.28 | 2.11 | 7.55 |
| | | Shot | 8.33 | 0.07 | 2.13 | 64.11 |
| | | Impulse | 4.76 | 0.31 | 4.30 | 1.80 |
| | | Speckle | 8.16 | 0.48 | 3.25 | 13.75 |
| | Blur | Defocus | 30.26 | 0.31 | 3.48 | 21.55 |
| | | Glass | 17.98 | 0.18 | 3.54 | 17.90 |
| | | Motion | 11.48 | 0.21 | 3.46 | 29.35 |
| | | Zoom | 5.40 | 0.14 | 6.71 | 2.70 |
| | Weather | Snow | 6.21 | 0.28 | 7.38 | 7.72 |
| | | Frost | 4.57 | 0.17 | 5.73 | 33.90 |
| | | Fog | 4.17 | 0.31 | 6.07 | 31.14 |
| | | Brightness | 3.41 | 0.28 | 3.80 | 1.19 |
| | Digital | Contrast | 1.18 | 0.08 | 1.96 | 3.88 |
| | | Elastic | 3.73 | 0.39 | 1.87 | 1.03 |
| | | Pixelate | 1.95 | 0.37 | 2.33 | 5.37 |
| | | JPEG | 9.94 | 0.23 | 3.24 | 4.80 |
| | Geomtric | Scale | 2.33 | 0.49 | 3.80 | 27.38 |
| | | Rotate | 5.66 | 0.07 | 2.37 | 0.27 |
| | | Shear | 3.85 | 0.05 | 2.09 | 3.41 |
| | | Piecewse Affine | 4.89 | 0.13 | 0.10 | 3.70 |

Table 6: Stable Diffusion image watermarks under image perturbations.

| Watermark | Attack Category | Perturbation | PSNR | SSIM | Bit Acc | Dect Acc |
|---|---|---|---|---|---|---|
| DctDwtSvd-WM | Noise | Gaussian | 19.12 | 0.39 | 1.80 | 16.93 |
| | | Shot | 18.62 | 0.37 | 2.08 | 25.33 |
| | | Impulse | 17.98 | 0.33 | 4.87 | 9.81 |
| | | Speckle | 20.22 | 0.49 | 3.73 | 59.36 |
| | Blur | Defocus | 23.93 | 0.69 | 11.01 | 35.49 |
| | | Glass | 27.61 | 0.84 | 15.26 | 93.41 |
| | | Motion | 21.20 | 0.66 | 7.77 | 8.46 |
| | | Zoom | 16.10 | 0.51 | 2.74 | 0.00 |
| | Weather | Snow | 11.36 | 0.48 | 0.37 | 5.71 |
| | | Frost | 16.44 | 0.72 | 6.68 | 16.89 |
| | | Fog | 12.66 | 0.60 | 0.55 | 0.84 |
| | | Brightness | 14.35 | 0.76 | 1.00 | 19.27 |
| | Digital | Contrast | 17.50 | 0.55 | 0.58 | 1.70 |
| | | Elastic | 21.05 | 0.49 | 6.94 | 14.85 |
| | | Pixelate | 22.03 | 0.55 | 8.06 | 0.56 |
| | | JPEG | 27.83 | 0.82 | 6.29 | 0.00 |
| | Geometric | Scale | 11.04 | 0.23 | 1.72 | 0.00 |
| | | Rotate | 9.23 | 0.17 | 1.27 | 0.00 |
| | | Shear | 10.96 | 0.24 | 2.03 | 0.00 |
| | | Piecewse Affine | 14.48 | 0.36 | 3.78 | 7.23 |

| Watermark | Attack Category | Perturbation | PSNR | SSIM | Bit Acc | Dect Acc |
|---|---|---|---|---|---|---|
| RivaGAN-WM | Noise | Gaussian | 19.03 | 0.35 | 1.78 | 12.69 |
| | | Shot | 18.56 | 0.37 | 2.00 | 15.65 |
| | | Impulse | 17.93 | 0.33 | 4.10 | 6.36 |
| | | Speckle | 20.14 | 0.49 | 3.07 | 29.42 |
| | Blur | Defocus | 23.81 | 0.69 | 7.00 | 46.98 |
| | | Glass | 27.23 | 0.82 | 7.91 | 95.22 |
| | | Motion | 21.14 | 0.65 | 5.34 | 47.78 |
| | | Zoom | 16.09 | 0.51 | 2.36 | 0.60 |
| | Weather | Snow | 11.38 | 0.48 | 0.37 | 6.85 |
| | | Frost | 15.47 | 0.71 | 1.53 | 46.23 |
| | | Fog | 12.58 | 0.60 | 0.56 | 8.27 |
| | | Brightness | 14.39 | 0.73 | 0.76 | 48.22 |
| | Digital | Contrast | 14.06 | 0.54 | 0.58 | 8.23 |
| | | Elastic | 17.46 | 0.49 | 3.62 | 84.18 |
| | | Pixelate | 21.01 | 0.55 | 5.51 | 4.17 |
| | | JPEG | 27.71 | 0.81 | 5.78 | 2.26 |
| | Geometric | Scale | 11.04 | 0.23 | 1.79 | 33.00 |
| | | Rotate | 9.23 | 0.17 | 1.30 | 13.00 |
| | | Shear | 10.96 | 0.24 | 2.31 | 60.28 |
| | | Piecewse Affine | 14.48 | 0.35 | 4.24 | 58.45 |

| Watermark | Attack Category | Perturbation | PSNR | SSIM | Bit Acc | Dect Acc |
|---|---|---|---|---|---|---|
| SSL-WM | Noise | Gaussian | 9.12 | 0.33 | 1.31 | 0.00 |
| | | Shot | 6.61 | 0.21 | 2.26 | 0.00 |
| | | Impulse | 7.03 | 0.51 | 4.83 | 0.00 |
| | | Speckle | 4.15 | 0.43 | 3.33 | 3.41 |
| | Blur | Defocus | 22.78 | 0.33 | 3.98 | 28.31 |
| | | Glass | 18.67 | 0.28 | 2.19 | 88.36 |
| | | Motion | 10.00 | 0.18 | 3.21 | 45.87 |
| | | Zoom | 9.53 | 0.27 | 1.19 | 4.86 |
| | Weather | Snow | 3.64 | 0.15 | 0.80 | 5.70 |
| | | Frost | 4.86 | 0.22 | 0.78 | 18.43 |
| | | Fog | 5.94 | 0.32 | 1.11 | 39.85 |
| | | Brightness | 6.93 | 0.33 | 1.41 | 41.83 |
| | Digital | Contrast | 6.92 | 0.38 | 2.85 | 65.78 |
| | | Elastic | 5.23 | 0.40 | 3.95 | 33.84 |
| | | Pixelate | 8.82 | 0.21 | 1.31 | 21.73 |
| | | JPEG | 7.98 | 0.38 | 4.43 | 22.38 |
| | Geometric | Scale | 5.79 | 0.38 | 2.36 | 24.46 |
| | | Rotate | 6.04 | 0.25 | 3.17 | 25.90 |
| | | Shear | 5.35 | 0.38 | 2.80 | 23.78 |
| | | Piecewse Affine | 6.28 | 0.18 | 2.88 | 0.00 |

| Watermark | Attack Category | Perturbation | PSNR | SSIM | Bit Acc | Dect Acc |
|---|---|---|---|---|---|---|
| StegaStamp-WM | Noise | Gaussian | 8.75 | 0.18 | 2.84 | 0.00 |
| | | Shot | 7.03 | 0.40 | 3.90 | 0.46 |
| | | Impulse | 6.31 | 0.38 | 3.34 | 0.37 |
| | | Speckle | 7.73 | 0.34 | 2.45 | 22.85 |
| | Blur | Defocus | 32.87 | 0.28 | 2.67 | 24.96 |
| | | Glass | 19.54 | 0.26 | 3.86 | 21.14 |
| | | Motion | 11.48 | 0.18 | 4.83 | 23.84 |
| | | Zoom | 4.95 | 0.00 | 5.28 | 14.80 |
| | Weather | Snow | 6.76 | 0.43 | 6.99 | 33.57 |
| | | Frost | 4.05 | 0.36 | 6.78 | 35.76 |
| | | Fog | 6.13 | 0.29 | 6.57 | 24.05 |
| | | Brightness | 1.23 | 0.35 | 6.47 | 0.00 |
| | Digital | Contrast | 1.13 | 0.22 | 6.78 | 0.00 |
| | | Elastic | 2.08 | 0.42 | 5.72 | 3.72 |
| | | Pixelate | 1.93 | 0.37 | 1.74 | 25.97 |
| | | JPEG | 9.28 | 0.25 | 1.85 | 33.78 |
| | Geomtric | Scale | 3.50 | 0.33 | 2.04 | 26.58 |
| | | Rotate | 4.82 | 0.37 | 2.38 | 38.56 |
| | | Shear | 3.81 | 0.22 | 2.17 | 27.31 |
| | | Piecewse Affine | 4.22 | 0.21 | 2.10 | 27.44 |

Table 7: DALLE3 image watermarks under image perturbations.

| Watermark | Attack Category | Perturbation | PSNR | SSIM | Bit Acc | Dect Acc |
|---|---|---|---|---|---|---|
| | | Gaussian | 18.91 | 0.39 | 1.83 | 12.61 |
| | Noise | Shot | 18.15 | 0.39 | 2.22 | 16.81 |
| | | Impulse | 18.18 | 0.37 | 4.42 | 7.38 |
| | | Speckle | 19.62 | 0.48 | 3.05 | 39.09 |
| | | Defocus | 22.90 | 0.64 | 5.64 | 33.88 |
| | Blur | Glass | 27.26 | 0.84 | 7.83 | 88.00 |
| | | Motion | 20.30 | 0.62 | 5.10 | 9.63 |
| | | Zoom | 16.18 | 0.49 | 2.83 | 0.00 |
| DctDwtSvd-WM | | Snow | 11.13 | 0.51 | 0.31 | 10.75 |
| | Weather | Frost | 10.36 | 0.60 | 0.40 | 5.13 |
| | | Fog | 13.21 | 0.61 | 0.65 | 1.08 |
| | | Brightness | 13.84 | 0.77 | 0.78 | 28.58 |
| | | Contrast | 14.48 | 0.54 | 0.83 | 2.40 |
| | Digital | Elastic | 16.50 | 0.43 | 4.20 | 17.17 |
| | | Pixelate | 21.04 | 0.56 | 5.23 | 17.97 |
| | | JPEG | 28.63 | 0.85 | 5.89 | 0.00 |
| | | Scale | 10.88 | 0.25 | 1.50 | 9.98 |
| | Geomtric | Rotate | 9.31 | 0.19 | 1.09 | 8.51 |
| | | Shear | 10.72 | 0.26 | 2.40 | 4.15 |
| | | Piecewse Affine | 15.50 | 0.43 | 5.58 | 3.09 |

| Watermark | Attack Category | Perturbation | PSNR | SSIM | Bit Acc | Dect Acc |
|---|---|---|---|---|---|---|
| | | Gaussian | 18.97 | 0.39 | 1.86 | 15.96 |
| | Noise | Shot | 18.21 | 0.40 | 2.33 | 14.4 |
| | | Impulse | 18.22 | 0.37 | 5.06 | 11.07 |
| | | Speckle | 19.71 | 0.49 | 3.71 | 23.89 |
| | | Defocus | 23.02 | 0.65 | 7.86 | 37.77 |
| | Blur | Glass | 27.64 | 0.86 | 12.50 | 91.53 |
| | | Motion | 20.34 | 0.62 | 6.30 | 38.05 |
| | | Zoom | 16.19 | 0.49 | 3.19 | 0.6 |
| RivaGAN-WM | | Snow | 11.17 | 0.52 | 0.32 | 5.9 |
| | Weather | Frost | 10.40 | 0.61 | 0.31 | 14.36 |
| | | Fog | 13.22 | 0.62 | 0.40 | 4.22 |
| | | Brightness | 13.98 | 0.79 | 0.65 | 56.86 |
| | | Contrast | 14.49 | 0.55 | 0.99 | 6.47 |
| | Digital | Elastic | 16.54 | 0.43 | 5.17 | 74.38 |
| | | Pixelate | 21.10 | 0.57 | 6.72 | 20.17 |
| | | JPEG | 28.82 | 0.85 | 6.20 | 0.55 |
| | | Scale | 10.87 | 0.25 | 1.36 | 75.57 |
| | Geomtric | Rotate | 9.31 | 0.18 | 1.00 | 16.16 |
| | | Shear | 10.81 | 0.27 | 1.81 | 0.00 |
| | | Piecewse Affine | 15.57 | 0.42 | 4.02 | 0.00 |

| Watermark | Attack Category | Perturbation | PSNR | SSIM | Bit Acc | Dect Acc |
|---|---|---|---|---|---|---|
| | | Gaussian | 10.77 | 0.02 | 0.08 | 3.41 |
| | Noise | Shot | 11.83 | 0.08 | 3.31 | 2.33 |
| | | Impulse | 5.58 | 0.19 | 3.76 | 4.82 |
| | | Speckle | 6.93 | 0.35 | 3.58 | 1.83 |
| | | Defocus | 22.73 | 0.28 | 3.17 | 18.88 |
| | Blur | Glass | 25.48 | 0.58 | 3.21 | 66.63 |
| | | Motion | 21.85 | 0.46 | 4.75 | 25.34 |
| | | Zoom | 18.42 | 0.31 | 4.24 | 28.64 |
| SSL-WM | | Snow | 4.31 | 0.27 | 3.82 | 27.80 |
| | Weather | Frost | 5.68 | 0.28 | 1.41 | 47.85 |
| | | Fog | 8.80 | 0.35 | 3.89 | 44.42 |
| | | Brightness | 9.41 | 0.34 | 5.91 | 41.23 |
| | | Contrast | 10.53 | 0.30 | 1.23 | 37.64 |
| | Digital | Elastic | 11.32 | 0.38 | 1.35 | 36.94 |
| | | Pixelate | 6.42 | 0.32 | 2.94 | 21.44 |
| | | JPEG | 3.31 | 0.35 | 3.74 | 22.85 |
| | | Scale | 5.83 | 0.14 | 9.44 | 38.74 |
| | Geomtric | Rotate | 2.78 | 0.32 | 1.42 | 14.84 |
| | | Shear | 3.33 | 0.02 | 2.83 | 24.00 |
| | | Piecewse Affine | 2.82 | 0.01 | 8.27 | 3.78 |

| Watermark | Attack Category | Perturbation | PSNR | SSIM | Bit Acc | Dect Acc |
|---|---|---|---|---|---|---|
| | | Gaussian | 5.02 | 0.24 | 1.49 | 54.41 |
| | Noise | Shot | 3.95 | 0.41 | 2.41 | 55.31 |
| | | Impulse | 7.33 | 0.11 | 2.01 | 2.25 |
| | | Speckle | 8.04 | 0.29 | 3.82 | 9.62 |
| | | Defocus | 8.75 | 0.20 | 6.93 | 17.83 |
| | Blur | Glass | 12.88 | 0.38 | 4.92 | 14.89 |
| | | Motion | 13.85 | 0.10 | 6.21 | 5.04 |
| | | Zoom | 13.00 | 0.17 | 4.82 | 1.73 |
| StegaStamp-WM | | Snow | 1.65 | 0.39 | 4.88 | 10.85 |
| | Weather | Frost | 3.28 | 0.28 | 6.28 | 29.59 |
| | | Fog | 3.90 | 0.12 | 2.08 | 18.13 |
| | | Brightness | 4.00 | 0.18 | 5.32 | 19.41 |
| | | Contrast | 8.53 | 0.42 | 0.53 | 27.74 |
| | Digital | Elastic | 20.41 | 0.40 | 3.88 | 24.93 |
| | | Pixelate | 3.45 | 0.43 | 5.83 | 1.07 |
| | | JPEG | 6.80 | 0.54 | 2.97 | 8.10 |
| | | Scale | 2.58 | 0.18 | 1.25 | 3.31 |
| | Geomtric | Rotate | 2.87 | 0.28 | 2.28 | 3.91 |
| | | Shear | 3.95 | 0.27 | 5.12 | 8.00 |
| | | Piecewse Affine | 2.37 | 0.39 | 3.76 | 3.41 |

Table 8: SDXL-Lightning image watermarks under image perturbations.

| Watermark | Attack Category | Perturbation | PSNR | SSIM | Bit Acc | Dect Acc |
|---|---|---|---|---|---|---|
| DctDwtSvd-WM | Noise | Gaussian | 18.90 | 0.29 | 1.55 | 19.75 |
| | | Shot | 18.08 | 0.28 | 1.52 | 22.86 |
| | | Impulse | 18.36 | 0.27 | 4.53 | 17.43 |
| | | Speckle | 19.73 | 0.38 | 2.50 | 51.08 |
| | Blur | Defocus | 26.79 | 0.79 | 17.35 | 38.88 |
| | | Glass | 31.14 | 0.91 | 23.44 | 96.75 |
| | | Motion | 23.46 | 0.74 | 11.21 | 14.25 |
| | | Zoom | 17.83 | 0.61 | 3.65 | 0.00 |
| | Weather | Snow | 11.14 | 0.49 | 0.03 | 16.47 |
| | | Frost | 16.97 | 0.72 | 7.08 | 25.29 |
| | | Fog | 13.71 | 0.69 | 0.69 | 0.80 |
| | | Brightness | 13.52 | 0.81 | 0.07 | 29.25 |
| | Digital | Contrast | 15.68 | 0.65 | 0.85 | 3.10 |
| | | Elastic | 19.63 | 0.58 | 10.55 | 26.39 |
| | | Pixelate | 23.30 | 0.65 | 12.28 | 7.29 |
| | | JPEG | 30.64 | 0.87 | 7.62 | 0.00 |
| | Geomtric | Scale | 12.46 | 0.35 | 1.81 | 22.00 |
| | | Rotate | 10.22 | 0.27 | 1.27 | 23.56 |
| | | Shear | 12.37 | 0.36 | 3.17 | 0.00 |
| | | Piecewse Affine | 17.06 | 0.52 | 8.02 | 10.17 |

| Watermark | Attack Category | Perturbation | PSNR | SSIM | Bit Acc | Dect Acc |
|---|---|---|---|---|---|---|
| RivaGAN-WM | Noise | Gaussian | 18.84 | 0.28 | 1.53 | 9.11 |
| | | Shot | 18.04 | 0.28 | 1.50 | 9.60 |
| | | Impulse | 18.31 | 0.27 | 3.92 | 5.23 |
| | | Speckle | 19.68 | 0.37 | 2.32 | 19.15 |
| | Blur | Defocus | 26.55 | 0.77 | 9.33 | 51.98 |
| | | Glass | 30.33 | 0.89 | 9.78 | 98.13 |
| | | Motion | 23.35 | 0.74 | 6.69 | 52.41 |
| | | Zoom | 17.82 | 0.60 | 3.18 | 0.98 |
| | Weather | Snow | 11.18 | 0.48 | 0.03 | 9.41 |
| | | Frost | 15.82 | 0.71 | 1.35 | 46.92 |
| | | Fog | 13.68 | 0.69 | 0.69 | 6.58 |
| | | Brightness | 13.62 | 0.77 | 0.08 | 54.60 |
| | Digital | Contrast | 15.67 | 0.65 | 0.85 | 6.15 |
| | | Elastic | 19.54 | 0.57 | 5.72 | 90.94 |
| | | Pixelate | 23.22 | 0.64 | 7.49 | 8.86 |
| | | JPEG | 30.28 | 0.86 | 6.95 | 1.90 |
| | Geomtric | Scale | 12.47 | 0.34 | 1.64 | 82.98 |
| | | Rotate | 10.22 | 0.26 | 1.15 | 16.29 |
| | | Shear | 12.37 | 0.35 | 2.13 | 76.05 |
| | | Piecewse Affine | 17.03 | 0.51 | 4.86 | 89.35 |

| Watermark | Attack Category | Perturbation | PSNR | SSIM | Bit Acc | Dect Acc |
|---|---|---|---|---|---|---|
| SSL-WM | Noise | Gaussian | 11.24 | 0.05 | 0.10 | 3.75 |
| | | Shot | 12.45 | 0.12 | 3.62 | 2.67 |
| | | Impulse | 6.12 | 0.24 | 3.87 | 5.16 |
| | | Speckle | 7.56 | 0.39 | 3.69 | 2.18 |
| | Blur | Defocus | 23.11 | 0.31 | 3.28 | 30.33 |
| | | Glass | 26.02 | 0.55 | 3.32 | 69.81 |
| | | Motion | 22.28 | 0.48 | 4.86 | 27.75 |
| | | Zoom | 19.05 | 0.33 | 4.35 | 31.06 |
| | Weather | Snow | 4.89 | 0.29 | 3.93 | 29.22 |
| | | Frost | 6.17 | 0.30 | 1.61 | 49.27 |
| | | Fog | 9.34 | 0.37 | 3.90 | 45.84 |
| | | Brightness | 10.03 | 0.36 | 6.02 | 42.65 |
| | Digital | Contrast | 11.15 | 0.32 | 1.43 | 38.06 |
| | | Elastic | 12.00 | 0.41 | 1.55 | 37.36 |
| | | Pixelate | 7.01 | 0.34 | 3.04 | 23.87 |
| | | JPEG | 3.87 | 0.37 | 3.84 | 25.28 |
| | Geomtric | Scale | 4.75 | 0.31 | 5.88 | 24.95 |
| | | Rotate | 6.27 | 0.28 | 4.72 | 24.24 |
| | | Shear | 5.43 | 0.30 | 4.70 | 28.57 |
| | | Piecewse Affine | 4.22 | 0.28 | 4.71 | 20.87 |

| Watermark | Attack Category | Perturbation | PSNR | SSIM | Bit Acc | Dect Acc |
|---|---|---|---|---|---|---|
| StegaStamp-WM | Noise | Gaussian | 5.19 | 0.53 | 9.75 | 9.11 |
| | | Shot | 3.85 | 0.05 | 2.57 | 2.84 |
| | | Impulse | 7.25 | 0.17 | 1.87 | 9.92 |
| | | Speckle | 8.61 | 0.24 | 3.95 | 14.80 |
| | Blur | Defocus | 6.48 | 0.28 | 7.01 | 1.20 |
| | | Glass | 12.11 | 0.38 | 5.02 | 1.17 |
| | | Motion | 9.03 | 0.31 | 4.18 | 3.10 |
| | | Zoom | 14.01 | 0.17 | 4.38 | 0.27 |
| | Weather | Snow | 2.78 | 0.19 | 6.28 | 1.50 |
| | | Frost | 3.48 | 0.02 | 6.82 | 14.04 |
| | | Fog | 4.13 | 0.03 | 2.47 | 29.84 |
| | | Brightness | 3.02 | 0.20 | 5.73 | 24.48 |
| | Digital | Contrast | 12.41 | 0.21 | 7.47 | 24.79 |
| | | Elastic | 16.82 | 0.15 | 2.75 | 21.28 |
| | | Pixelate | 3.57 | 0.24 | 4.72 | 5.34 |
| | | JPEG | 6.63 | 0.22 | 5.37 | 4.32 |
| | Geomtric | Scale | 2.73 | 0.38 | 3.88 | 3.57 |
| | | Rotate | 4.93 | 0.42 | 1.57 | 5.16 |
| | | Shear | 2.54 | 0.21 | 4.38 | 6.28 |
| | | Piecewse Affine | 2.44 | 0.10 | 5.07 | 2.04 |

Table 9: PIXART image watermarks under image perturbations.

| Watermark | Attack Category | Perturbation | PSNR | SSIM | Bit Acc | Dect Acc |
|---|---|---|---|---|---|---|
| DctDwtSvd-WM | Noise | Gaussian | 19.19 | 0.30 | 2.23 | 14.15 |
| | | Shot | 19.23 | 0.36 | 3.40 | 35.73 |
| | | Impulse | 17.70 | 0.27 | 5.01 | 8.26 |
| | | Speckle | 20.87 | 0.49 | 4.75 | 67.80 |
| | Blur | Defocus | 24.43 | 0.75 | 8.24 | 38.18 |
| | | Glass | 27.98 | 0.86 | 10.37 | 92.87 |
| | | Motion | 21.89 | 0.71 | 7.40 | 9.56 |
| | | Zoom | 16.86 | 0.57 | 3.47 | 0.00 |
| | Weather | Snow | 11.45 | 0.43 | 0.49 | 10.60 |
| | | Frost | 15.81 | 0.66 | 2.68 | 19.44 |
| | | Fog | 11.88 | 0.57 | 0.47 | 0.72 |
| | | Brightness | 14.77 | 0.70 | 1.69 | 18.32 |
| | Digital | Contrast | 13.94 | 0.56 | 0.61 | 2.10 |
| | | Elastic | 18.28 | 0.57 | 6.32 | 21.72 |
| | | Pixelate | 21.65 | 0.55 | 7.07 | 5.11 |
| | | JPEG | 28.70 | 0.83 | 7.23 | 0.00 |
| | Geomtric | Scale | 11.36 | 0.33 | 2.26 | 1.43 |
| | | Rotate | 9.83 | 0.26 | 2.06 | 16.60 |
| | | Shear | 11.18 | 0.35 | 3.05 | 6.49 |
| | | Piecewse Affine | 16.28 | 0.51 | 6.80 | 7.56 |

| Watermark | Attack Category | Perturbation | PSNR | SSIM | Bit Acc | Dect Acc |
|---|---|---|---|---|---|---|
| RivaGAN-WM | Noise | Gaussian | 19.25 | 0.30 | 2.27 | 5.58 |
| | | Shot | 19.31 | 0.37 | 3.79 | 13.11 |
| | | Impulse | 17.73 | 0.27 | 5.76 | 1.81 |
| | | Speckle | 20.98 | 0.51 | 7.90 | 28.58 |
| | Blur | Defocus | 24.58 | 0.76 | 16.53 | 47.98 |
| | | Glass | 28.39 | 0.89 | 22.61 | 94.15 |
| | | Motion | 21.96 | 0.73 | 12.57 | 48.42 |
| | | Zoom | 16.87 | 0.58 | 5.23 | 0.47 |
| | Weather | Snow | 11.50 | 0.44 | 0.50 | 5.56 |
| | | Frost | 16.50 | 0.67 | 7.89 | 42.01 |
| | | Fog | 11.88 | 0.57 | 0.46 | 4.85 |
| | | Brightness | 14.97 | 0.73 | 2.27 | 31.75 |
| | Digital | Contrast | 13.94 | 0.56 | 0.61 | 5.60 |
| | | Elastic | 18.30 | 0.58 | 11.13 | 82.63 |
| | | Pixelate | 21.70 | 0.64 | 12.52 | 5.54 |
| | | JPEG | 28.93 | 0.84 | 7.92 | 2.21 |
| | Geomtric | Scale | 11.36 | 0.33 | 3.15 | 74.49 |
| | | Rotate | 9.83 | 0.26 | 2.73 | 15.30 |
| | | Shear | 11.18 | 0.35 | 4.96 | 69.68 |
| | | Piecewse Affine | 17.27 | 0.52 | 9.30 | 82.23 |

| Watermark | Attack Category | Perturbation | PSNR | SSIM | Bit Acc | Dect Acc |
|---|---|---|---|---|---|---|
| SSL-WM | Noise | Gaussian | 9.56 | 0.35 | 1.43 | 0.31 |
| | | Shot | 7.07 | 0.23 | 2.38 | 0.99 |
| | | Impulse | 7.49 | 0.52 | 4.95 | 0.45 |
| | | Speckle | 4.31 | 0.44 | 3.45 | 3.61 |
| | Blur | Defocus | 23.04 | 0.34 | 4.10 | 39.60 |
| | | Glass | 19.93 | 0.29 | 2.21 | 89.56 |
| | | Motion | 11.30 | 0.19 | 3.23 | 46.95 |
| | | Zoom | 10.83 | 0.27 | 1.29 | 5.16 |
| | Weather | Snow | 4.14 | 0.16 | 0.82 | 6.00 |
| | | Frost | 5.36 | 0.23 | 0.80 | 19.53 |
| | | Fog | 6.44 | 0.33 | 1.13 | 39.45 |
| | | Brightness | 7.43 | 0.34 | 1.43 | 42.13 |
| | Digital | Contrast | 7.42 | 0.49 | 2.87 | 55.06 |
| | | Elastic | 5.73 | 0.41 | 3.98 | 34.14 |
| | | Pixelate | 9.02 | 0.22 | 1.33 | 22.93 |
| | | JPEG | 8.18 | 0.39 | 4.45 | 23.56 |
| | Geomtric | Scale | 9.90 | 0.02 | 4.04 | 23.46 |
| | | Rotate | 7.65 | 0.37 | 5.73 | 22.74 |
| | | Shear | 2.64 | 0.28 | 6.79 | 21.57 |
| | | Piecewse Affine | 3.35 | 0.21 | 5.46 | 0.00 |

| Watermark | Attack Category | Perturbation | PSNR | SSIM | Bit Acc | Dect Acc |
|---|---|---|---|---|---|---|
| StegaStamp-WM | Noise | Gaussian | 4.07 | 0.24 | 2.11 | 21.84 |
| | | Shot | 4.47 | 0.53 | 2.49 | 6.72 |
| | | Impulse | 7.58 | 0.24 | 1.92 | 14.95 |
| | | Speckle | 8.58 | 0.16 | 3.00 | 18.75 |
| | Blur | Defocus | 10.33 | 0.39 | 4.82 | 21.75 |
| | | Glass | 11.74 | 0.05 | 5.08 | 22.92 |
| | | Motion | 13.75 | 0.42 | 5.71 | 10.32 |
| | | Zoom | 10.99 | 0.00 | 4.32 | 20.30 |
| | Weather | Snow | 10.41 | 0.45 | 5.55 | 44.78 |
| | | Frost | 17.04 | 0.37 | 4.18 | 17.81 |
| | | Fog | 3.56 | 0.03 | 2.25 | 14.75 |
| | | Brightness | 4.08 | 0.10 | 5.14 | 21.80 |
| | Digital | Contrast | 13.77 | 0.09 | 0.05 | 23.72 |
| | | Elastic | 18.80 | 0.22 | 2.10 | 3.11 |
| | | Pixelate | 3.53 | 0.21 | 3.04 | 1.82 |
| | | JPEG | 6.10 | 0.30 | 5.28 | 9.72 |
| | Geomtric | Scale | 7.75 | 0.04 | 3.48 | 1.85 |
| | | Rotate | 1.78 | 0.09 | 0.24 | 5.60 |
| | | Shear | 4.37 | 0.05 | 2.93 | 4.42 |
| | | Piecewse Affine | 0.28 | 0.13 | 2.47 | 3.72 |

Table 10: Kandinsky 2.2 image watermarks under image perturbations.

| Watermark | Attack Category | Perturbation | PSNR | SSIM | Bit Acc | Dect Acc |
|---|---|---|---|---|---|---|
| DctDwtSvd-WM | Noise | Gaussian | 19.25 | 0.46 | 2.90 | 1.20 |
| | | Shot | 18.71 | 0.49 | 4.41 | 7.82 |
| | | Impulse | 17.55 | 0.42 | 6.01 | 0.12 |
| | | Speckle | 19.64 | 0.58 | 5.44 | 22.25 |
| | Blur | Defocus | 21.15 | 0.64 | 5.96 | 19.12 |
| | | Glass | 25.33 | 0.83 | 8.45 | 58.70 |
| | | Motion | 18.63 | 0.60 | 5.44 | 2.37 |
| | | Zoom | 15.09 | 0.45 | 3.27 | 0.00 |
| | Weather | Snow | 11.65 | 0.53 | 1.46 | 0.66 |
| | | Frost | 10.25 | 0.54 | 1.47 | 0.40 |
| | | Fog | 11.27 | 0.50 | 0.39 | 0.08 |
| | | Brightness | 15.85 | 0.72 | 3.88 | 10.37 |
| | Digital | Contrast | 12.48 | 0.44 | 0.41 | 0.72 |
| | | Elastic | 14.65 | 0.40 | 4.78 | 4.81 |
| | | Pixelate | 20.95 | 0.65 | 6.32 | 20.45 |
| | | JPEG | 26.83 | 0.84 | 6.89 | 0.00 |
| | Geomtric | Scale | 9.85 | 0.19 | 2.57 | 0.00 |
| | | Rotate | 8.15 | 0.14 | 2.50 | 0.00 |
| | | Shear | 9.91 | 0.23 | 3.20 | 0.00 |
| | | Piecewse Affine | 12.70 | 0.32 | 4.43 | 1.13 |

| Watermark | Attack Category | Perturbation | PSNR | SSIM | Bit Acc | Dect Acc |
|---|---|---|---|---|---|---|
| RivaGAN-WM | Noise | Gaussian | 19.31 | 0.46 | 3.00 | 7.05 |
| | | Shot | 18.77 | 0.50 | 4.78 | 9.32 |
| | | Impulse | 17.59 | 0.42 | 6.82 | 3.17 |
| | | Speckle | 19.72 | 0.59 | 7.89 | 16.15 |
| | Blur | Defocus | 21.23 | 0.65 | 9.64 | 25.81 |
| | | Glass | 25.57 | 0.85 | 14.67 | 68.94 |
| | | Motion | 18.65 | 0.61 | 7.86 | 24.75 |
| | | Zoom | 15.10 | 0.46 | 4.76 | 1.90 |
| | Weather | Snow | 11.69 | 0.53 | 1.48 | 2.33 |
| | | Frost | 10.29 | 0.55 | 1.47 | 5.24 |
| | | Fog | 11.29 | 0.50 | 0.40 | 1.13 |
| | | Brightness | 16.05 | 0.74 | 5.24 | 32.11 |
| | Digital | Contrast | 12.49 | 0.44 | 0.41 | 3.43 |
| | | Elastic | 14.65 | 0.40 | 7.17 | 40.7 |
| | | Pixelate | 21.02 | 0.67 | 10.42 | 32.35 |
| | | JPEG | 27.00 | 0.85 | 7.29 | 0.4 |
| | Geomtric | Scale | 9.86 | 0.19 | 3.36 | 3.86 |
| | | Rotate | 8.15 | 0.14 | 3.03 | 4.85 |
| | | Shear | 9.91 | 0.23 | 4.68 | 40.43 |
| | | Piecewse Affine | 12.69 | 0.33 | 6.13 | 19.56 |

| Watermark | Attack Category | Perturbation | PSNR | SSIM | Bit Acc | Dect Acc |
|---|---|---|---|---|---|---|
| SSL-WM | Noise | Gaussian | 5.82 | 0.15 | 1.19 | 1.11 |
| | | Shot | 6.88 | 0.22 | 2.83 | 1.41 |
| | | Impulse | 7.31 | 0.31 | 3.33 | 21.34 |
| | | Speckle | 7.82 | 0.10 | 6.21 | 26.59 |
| | Blur | Defocus | 7.95 | 0.11 | 4.31 | 27.46 |
| | | Glass | 10.31 | 0.51 | 5.73 | 88.21 |
| | | Motion | 11.36 | 0.44 | 4.63 | 15.74 |
| | | Zoom | 14.85 | 0.38 | 4.89 | 3.15 |
| | Weather | Snow | 7.52 | 0.39 | 5.81 | 2.22 |
| | | Frost | 8.32 | 0.26 | 3.26 | 45.64 |
| | | Fog | 6.54 | 0.22 | 4.71 | 33.75 |
| | | Brightness | 7.83 | 0.25 | 2.84 | 32.85 |
| | Digital | Contrast | 11.41 | 0.21 | 3.91 | 29.58 |
| | | Elastic | 10.88 | 0.22 | 2.74 | 27.04 |
| | | Pixelate | 9.93 | 0.16 | 3.85 | 0.00 |
| | | JPEG | 9.40 | 0.19 | 2.41 | 0.00 |
| | Geomtric | Scale | 8.26 | 0.15 | 4.05 | 14.88 |
| | | Rotate | 5.81 | 0.18 | 3.96 | 15.96 |
| | | Shear | 6.61 | 0.17 | 2.78 | 11.21 |
| | | Piecewse Affine | 4.48 | 0.11 | 4.57 | 20.07 |

| Watermark | Attack Category | Perturbation | PSNR | SSIM | Bit Acc | Dect Acc |
|---|---|---|---|---|---|---|
| StegaStamp-WM | Noise | Gaussian | 4.33 | 0.12 | 1.12 | 1.00 |
| | | Shot | 4.29 | 0.20 | 2.04 | 1.04 |
| | | Impulse | 6.12 | 0.14 | 2.58 | 10.33 |
| | | Speckle | 7.87 | 0.12 | 3.41 | 15.89 |
| | Blur | Defocus | 7.71 | 0.18 | 5.55 | 16.45 |
| | | Glass | 12.44 | 0.43 | 5.89 | 12.44 |
| | | Motion | 13.28 | 0.28 | 5.76 | 3.33 |
| | | Zoom | 13.77 | 0.33 | 5.04 | 1.04 |
| | Weather | Snow | 1.19 | 0.37 | 5.94 | 5.58 |
| | | Frost | 3.21 | 0.30 | 5.89 | 22.89 |
| | | Fog | 3.91 | 0.29 | 3.20 | 25.75 |
| | | Brightness | 4.22 | 0.22 | 4.60 | 23.06 |
| | Digital | Contrast | 12.87 | 0.24 | 1.05 | 24.85 |
| | | Elastic | 19.30 | 0.21 | 2.00 | 20.00 |
| | | Pixelate | 3.34 | 0.19 | 3.85 | 1.41 |
| | | JPEG | 5.69 | 0.18 | 4.58 | 1.63 |
| | Geomtric | Scale | 6.38 | 0.16 | 3.88 | 0.00 |
| | | Rotate | 2.48 | 0.10 | 1.03 | 0.00 |
| | | Shear | 3.90 | 0.15 | 2.43 | 2.24 |
| | | Piecewse Affine | 1.86 | 0.15 | 3.85 | 3.90 |

Table 11: LCMs image watermarks under image perturbations.

| Watermark | Attack Category | Perturbation | PSNR | SSIM | Bit Acc | Dect Acc |
|---|---|---|---|---|---|---|
| DctDwtSvd-WM | Noise | Gaussian | 18.85 | 0.24 | 1.53 | 19.66 |
| | | Shot | 17.81 | 0.23 | 1.44 | 20.34 |
| | | Impulse | 18.46 | 0.23 | 4.51 | 18.28 |
| | | Speckle | 19.34 | 0.31 | 2.24 | 44.70 |
| | Blur | Defocus | 29.37 | 0.81 | 18.77 | 39.10 |
| | | Glass | 33.14 | 0.91 | 24.74 | 96.58 |
| | | Motion | 25.81 | 0.78 | 12.02 | 14.33 |
| | | Zoom | 19.29 | 0.67 | 4.27 | 0.00 |
| | Weather | Snow | 11.13 | 0.49 | 0.01 | 12.71 |
| | | Frost | 17.19 | 0.73 | 7.50 | 24.38 |
| | | Fog | 14.03 | 0.73 | 0.72 | 1.22 |
| | | Brightness | 13.56 | 0.83 | 0.02 | 38.52 |
| | Digital | Contrast | 16.35 | 0.70 | 0.97 | 3.04 |
| | | Elastic | 21.93 | 0.64 | 11.24 | 26.81 |
| | | Pixelate | 25.76 | 0.69 | 13.33 | 6.41 |
| | | JPEG | 31.18 | 0.85 | 7.85 | 0.00 |
| | Geomtric | Scale | 13.68 | 0.43 | 2.07 | 0.00 |
| | | Rotate | 10.54 | 0.33 | 1.37 | 0.00 |
| | | Shear | 13.62 | 0.43 | 3.41 | 0.00 |
| | | Piecewse Affine | 18.48 | 0.58 | 8.48 | 10.05 |

| Watermark | Attack Category | Perturbation | PSNR | SSIM | Bit Acc | Dect Acc |
|---|---|---|---|---|---|---|
| RivaGAN-WM | Noise | Gaussian | 18.78 | 0.24 | 1.51 | 7.39 |
| | | Shot | 17.77 | 0.22 | 1.42 | 6.15 |
| | | Impulse | 18.41 | 0.23 | 3.90 | 3.88 |
| | | Speckle | 19.29 | 0.31 | 2.09 | 13.59 |
| | Blur | Defocus | 28.94 | 0.80 | 10.22 | 51.71 |
| | | Glass | 31.89 | 0.88 | 10.28 | 97.60 |
| | | Motion | 25.58 | 0.77 | 7.46 | 52.01 |
| | | Zoom | 19.28 | 0.66 | 3.77 | 1.18 |
| | Weather | Snow | 11.17 | 0.48 | 0.01 | 6.57 |
| | | Frost | 15.79 | 0.71 | 1.31 | 43.64 |
| | | Fog | 14.04 | 0.72 | 0.73 | 6.07 |
| | | Brightness | 13.65 | 0.79 | 0.04 | 52.79 |
| | Digital | Contrast | 16.34 | 0.69 | 0.97 | 6.07 |
| | | Elastic | 21.86 | 0.63 | 6.27 | 91.13 |
| | | Pixelate | 25.63 | 0.69 | 8.44 | 9.75 |
| | | JPEG | 30.78 | 0.84 | 7.19 | 1.32 |
| | Geomtric | Scale | 13.68 | 0.42 | 1.92 | 84.81 |
| | | Rotate | 10.54 | 0.33 | 1.29 | 16.27 |
| | | Shear | 13.63 | 0.43 | 2.45 | 75.16 |
| | | Piecewse Affine | 18.46 | 0.57 | 5.35 | 89.71 |

| Watermark | Attack Category | Perturbation | PSNR | SSIM | Bit Acc | Dect Acc |
|---|---|---|---|---|---|---|
| SSL-WM | Noise | Gaussian | 9.62 | 0.38 | 1.45 | 0.42 |
| | | Shot | 7.13 | 0.28 | 2.40 | 0.75 |
| | | Impulse | 7.55 | 0.75 | 3.99 | 0.57 |
| | | Speckle | 4.46 | 0.45 | 3.45 | 1.52 |
| | Blur | Defocus | 23.25 | 0.38 | 4.13 | 18.52 |
| | | Glass | 20.15 | 0.28 | 2.25 | 3.58 |
| | | Motion | 11.52 | 0.20 | 3.50 | 5.23 |
| | | Zoom | 10.95 | 0.28 | 1.55 | 6.93 |
| | Weather | Snow | 4.32 | 0.11 | 0.66 | 8.76 |
| | | Frost | 5.48 | 0.28 | 0.84 | 11.03 |
| | | Fog | 6.32 | 0.48 | 1.24 | 20.41 |
| | | Brightness | 6.12 | 0.38 | 1.56 | 44.21 |
| | Digital | Contrast | 6.84 | 0.50 | 3.56 | 0.41 |
| | | Elastic | 5.85 | 0.48 | 4.10 | 92.45 |
| | | Pixelate | 9.16 | 0.11 | 1.46 | 3.31 |
| | | JPEG | 8.28 | 0.48 | 4.50 | 2.54 |
| | Geomtric | Scale | 9.84 | 0.33 | 3.94 | 1.15 |
| | | Rotate | 8.48 | 0.26 | 6.37 | 7.88 |
| | | Shear | 2.04 | 0.24 | 4.76 | 5.47 |
| | | Piecewse Affine | 7.64 | 0.15 | 5.47 | 5.10 |

| Watermark | Attack Category | Perturbation | PSNR | SSIM | Bit Acc | Dect Acc |
|---|---|---|---|---|---|---|
| StegaStamp-WM | Noise | Gaussian | 2.91 | 0.16 | 12.84 | 18.82 |
| | | Shot | 4.92 | 0.04 | 2.74 | 0.94 |
| | | Impulse | 5.46 | 0.14 | 2.52 | 2.89 |
| | | Speckle | 7.68 | 0.13 | 3.48 | 25.13 |
| | Blur | Defocus | 8.74 | 0.33 | 5.34 | 22.31 |
| | | Glass | 12.66 | 0.17 | 5.40 | 11.38 |
| | | Motion | 14.49 | 0.10 | 6.18 | 5.78 |
| | | Zoom | 12.15 | 0.21 | 4.24 | 9.45 |
| | Weather | Snow | 0.49 | 0.19 | 6.84 | 0.21 |
| | | Frost | 3.82 | 0.28 | 4.85 | 20.30 |
| | | Fog | 3.46 | 0.49 | 1.20 | 25.19 |
| | | Brightness | 5.60 | 0.04 | 3.28 | 29.80 |
| | Digital | Contrast | 13.30 | 0.24 | 3.45 | 25.79 |
| | | Elastic | 18.82 | 0.30 | 0.04 | 21.78 |
| | | Pixelate | 5.00 | 0.18 | 4.14 | 6.78 |
| | | JPEG | 3.92 | 0.21 | 3.88 | 6.82 |
| | Geomtric | Scale | 6.84 | 0.40 | 4.94 | 0.07 |
| | | Rotate | 2.83 | 0.21 | 1.98 | 3.45 |
| | | Shear | 4.55 | 0.12 | 2.72 | 11.42 |
| | | Piecewse Affine | 1.12 | 0.30 | 2.88 | 3.98 |

Table 12: RPG image watermarks under image perturbations.

| Watermark | Attack Category | Perturbation | PSNR | SSIM | Bit Acc | Dect Acc |
|---|---|---|---|---|---|---|
| DctDwtSvd-WM | Noise | Gaussian | 21.21 | 0.53 | 4.42 | 38.39 |
| | | Shot | 17.95 | 0.45 | 2.97 | 23.52 |
| | | Impulse | 21.83 | 0.58 | 4.74 | 39.05 |
| | | Speckle | 26.15 | 0.73 | 6.71 | 56.49 |
| | Blur | Defocus | 17.25 | 0.48 | 2.72 | 17.63 |
| | | Glass | 17.17 | 0.50 | 2.70 | 16.35 |
| | | Motion | 26.21 | 0.78 | 6.80 | 53.85 |
| | | Zoom | 22.06 | 0.67 | 4.94 | 35.21 |
| | Weather | Snow | 15.77 | 0.51 | 2.13 | 7.47 |
| | | Frost | 20.78 | 0.67 | 4.41 | 27.84 |
| | | Fog | 15.66 | 0.54 | 2.12 | 5.04 |
| | | Brightness | 15.57 | 0.55 | 2.10 | 3.71 |
| | Digital | Contrast | 19.05 | 0.67 | 3.69 | 17.57 |
| | | Elastic | 8.15 | 0.37 | 1.20 | 29.81 |
| | | Pixelate | 9.02 | 0.41 | 0.79 | 27.06 |
| | | JPEG | 14.79 | 0.60 | 1.83 | 3.48 |
| | Geomtric | Scale | 22.48 | 0.14 | 15.02 | 25.19 |
| | | Rotate | 15.64 | 0.29 | 1.19 | 30.92 |
| | | Shear | 14.58 | 0.28 | 4.05 | 55.83 |
| | | Piecewse Affine | 13.57 | 0.24 | 1.29 | 69.31 |

| Watermark | Attack Category | Perturbation | PSNR | SSIM | Bit Acc | Dect Acc |
|---|---|---|---|---|---|---|
| RivaGAN-WM | Noise | Gaussian | 21.33 | 0.54 | 5.98 | 38.89 |
| | | Shot | 18.03 | 0.46 | 3.83 | 22.22 |
| | | Impulse | 21.95 | 0.59 | 6.39 | 42.90 |
| | | Speckle | 26.34 | 0.74 | 9.24 | 65.93 |
| | Blur | Defocus | 17.32 | 0.49 | 3.37 | 19.77 |
| | | Glass | 17.24 | 0.50 | 3.32 | 19.80 |
| | | Motion | 26.40 | 0.79 | 9.28 | 67.48 |
| | | Zoom | 22.19 | 0.68 | 6.54 | 46.17 |
| | Weather | Snow | 15.83 | 0.51 | 2.39 | 13.68 |
| | | Frost | 20.90 | 0.68 | 5.70 | 40.31 |
| | | Fog | 15.71 | 0.54 | 2.32 | 13.89 |
| | | Brightness | 15.62 | 0.56 | 2.26 | 13.85 |
| | Digital | Contrast | 19.15 | 0.68 | 4.56 | 32.48 |
| | | Elastic | 8.11 | 0.37 | 2.63 | 24.13 |
| | | Pixelate | 8.99 | 0.41 | 2.06 | 19.16 |
| | | JPEG | 14.83 | 0.60 | 1.75 | 11.41 |
| | Geomtric | Scale | 15.72 | 0.24 | 4.45 | 12.42 |
| | | Rotate | 12.68 | 0.28 | 3.72 | 27.98 |
| | | Shear | 13.66 | 0.18 | 2.18 | 25.54 |
| | | Piecewse Affine | 14.03 | 0.25 | 3.79 | 32.88 |

| Watermark | Attack Category | Perturbation | PSNR | SSIM | Bit Acc | Dect Acc |
|---|---|---|---|---|---|---|
| SSL-WM | Noise | Gaussian | 18.40 | 0.27 | 3.70 | 21.26 |
| | | Shot | 13.65 | 0.21 | 2.81 | 11.83 |
| | | Impulse | 18.45 | 0.32 | 3.96 | 27.87 |
| | | Speckle | 23.86 | 0.44 | 5.23 | 45.50 |
| | Blur | Defocus | 11.56 | 0.23 | 2.74 | 15.97 |
| | | Glass | 11.08 | 0.24 | 2.76 | 17.93 |
| | | Motion | 22.80 | 0.49 | 5.38 | 52.36 |
| | | Zoom | 16.86 | 0.40 | 4.24 | 39.77 |
| | Weather | Snow | 8.06 | 0.25 | 2.49 | 19.57 |
| | | Frost | 14.39 | 0.39 | 3.96 | 39.65 |
| | | Fog | 7.14 | 0.28 | 2.54 | 23.58 |
| | | Brightness | 6.65 | 0.29 | 2.56 | 25.49 |
| | Digital | Contrast | 10.93 | 0.39 | 3.59 | 40.12 |
| | | Elastic | 4.05 | 0.13 | 0.53 | 3.47 |
| | | Pixelate | 3.26 | 0.16 | 0.82 | 8.80 |
| | | JPEG | 4.08 | 0.32 | 2.51 | 31.58 |
| | Geomtric | Scale | 3.44 | 0.33 | 5.85 | 26.35 |
| | | Rotate | 3.85 | 0.30 | 4.07 | 22.81 |
| | | Shear | 2.49 | 0.18 | 4.72 | 0.00 |
| | | Piecewse Affine | 5.84 | 0.29 | 3.47 | 0.00 |

| Watermark | Attack Category | Perturbation | PSNR | SSIM | Bit Acc | Dect Acc |
|---|---|---|---|---|---|---|
| StegaStamp-WM | Noise | Gaussian | 5.66 | 0.49 | 1.89 | 0.00 |
| | | Shot | 6.00 | 0.10 | 1.50 | 0.00 |
| | | Impulse | 6.92 | 0.15 | 3.45 | 0.00 |
| | | Speckle | 8.00 | 0.19 | 4.49 | 15.50 |
| | Blur | Defocus | 7.88 | 0.32 | 4.41 | 19.93 |
| | | Glass | 12.30 | 0.33 | 4.90 | 11.98 |
| | | Motion | 12.78 | 0.38 | 5.98 | 4.19 |
| | | Zoom | 13.42 | 0.12 | 4.43 | 4.28 |
| | Weather | Snow | 3.42 | 0.07 | 4.29 | 2.42 |
| | | Frost | 2.04 | 0.42 | 4.79 | 18.14 |
| | | Fog | 3.28 | 0.13 | 3.48 | 2.83 |
| | | Brightness | 5.52 | 0.41 | 4.48 | 21.48 |
| | Digital | Contrast | 10.92 | 0.01 | 1.12 | 22.48 |
| | | Elastic | 0.32 | 0.28 | 2.50 | 24.39 |
| | | Pixelate | 1.82 | 0.44 | 3.45 | 7.58 |
| | | JPEG | 4.18 | 0.14 | 2.58 | 4.92 |
| | Geomtric | Scale | 4.30 | 0.32 | 5.29 | 2.40 |
| | | Rotate | 1.83 | 0.24 | 2.71 | 3.39 |
| | | Shear | 4.23 | 0.47 | 0.41 | 5.28 |
| | | Piecewse Affine | 1.43 | 0.19 | 3.15 | 3.58 |

Table 13: NExT-GPT text watermarks under text perturbations.

| Watermark | Attack Category | Perturbation | BLEURT | ROUGE | Bit Acc | Dect Acc |
|---|---|---|---|---|---|---|
| KGW-WM | Character-level | Keyboard | 0.40 | 57.96 | 65.00 | 17.70 |
| | | OCR | 0.58 | 74.09 | 65.22 | 17.01 |
| | | Character Insert (CI) | 0.42 | 58.15 | 65.77 | 18.37 |
| | | Character Replace (CR) | 0.40 | 58.15 | 64.82 | 18.37 |
| | | Character Swap (CS) | 0.51 | 61.40 | 68.17 | 19.64 |
| | | Character Delete (CD) | 0.47 | 58.17 | 66.30 | 15.09 |
| | Word-level | Synonym Replacement (SR) | 0.54 | 62.54 | 68.48 | 24.83 |
| | | Word Insertion (WI) | 0.53 | 76.60 | 64.02 | 25.87 |
| | | Word Swap (WS) | 0.54 | 59.88 | 67.62 | 31.01 |
| | | Word Deletion (WD) | 0.50 | 72.92 | 63.66 | 26.51 |
| | | Insert Punctuation (IP) | 0.06 | 6.99 | 53.77 | 26.49 |
| | Sentence-level | Formal | 0.02 | 0.46 | 50.43 | 17.72 |
| | | Casual | 0.01 | 2.44 | 50.28 | 18.83 |
| | | Passive | 0.07 | 7.43 | 53.74 | 17.07 |
| | | Active | 0.07 | 8.16 | 54.86 | 14.88 |
| | | Back Translation | 0.79 | 77.41 | 77.96 | 17.03 |
| | | SCPN | 0.09 | 2.15 | 44.82 | 11.31 |
| | | Bart | 0.00 | 3.66 | 41.06 | 16.74 |
| | | Dipper | 0.04 | 2.75 | 42.78 | 17.89 |

| Watermark | Attack Category | Perturbation | BLEURT | ROUGE | Bit Acc | Dect Acc |
|---|---|---|---|---|---|---|
| KTH-WM | Character-level | Keyboard | 0.39 | 63.88 | 64.88 | 18.67 |
| | | OCR | 0.56 | 77.77 | 65.15 | 19.43 |
| | | Character Insert (CI) | 0.41 | 64.02 | 64.79 | 19.54 |
| | | Character Replace (CR) | 0.39 | 64.04 | 64.74 | 19.60 |
| | | Character Swap (CS) | 0.49 | 66.66 | 66.43 | 22.99 |
| | | Character Delete (CD) | 0.46 | 64.17 | 65.72 | 17.90 |
| | Word-level | Synonym Replacement (SR) | 0.44 | 53.98 | 66.24 | 26.73 |
| | | Word Insertion (WI) | 0.45 | 70.53 | 63.42 | 28.01 |
| | | Word Swap (WS) | 0.47 | 56.77 | 65.67 | 30.01 |
| | | Word Deletion (WD) | 0.44 | 68.20 | 63.13 | 27.60 |
| | | Insert Punctuation (IP) | 0.17 | 4.83 | 54.08 | 29.04 |
| | Sentence-level | Formal | 0.05 | 0.19 | 17.69 | 9.08 |
| | | Casual | 0.04 | 0.52 | 14.61 | 6.63 |
| | | Passive | 0.18 | 2.88 | 55.40 | 25.02 |
| | | Active | 0.19 | 3.03 | 55.40 | 23.52 |
| | | Back Translation | 0.64 | 72.46 | 71.33 | 23.19 |
| | | SCPN | 0.00 | 23.62 | 20.89 | 19.70 |
| | | Bart | 0.02 | 13.77 | 21.75 | 18.64 |
| | | Dipper | 0.06 | 15.84 | 22.57 | 14.75 |

| Watermark | Attack Category | Perturbation | BLEURT | ROUGE | Bit Acc | Dect Acc |
|---|---|---|---|---|---|---|
| Blackbox-WM | Character-level | Keyboard | 0.42 | 65.78 | 81.13 | 41.82 |
| | | OCR | 0.59 | 79.28 | 82.75 | 50.98 |
| | | Character Insert (CI) | 0.44 | 66.00 | 68.85 | 40.73 |
| | | Character Replace (CR) | 0.42 | 66.00 | 81.24 | 41.02 |
| | | Character Swap (CS) | 0.53 | 68.98 | 81.33 | 42.39 |
| | | Character Delete (CD) | 0.48 | 66.03 | 69.65 | 39.87 |
| | Word-level | Synonym Replacement (SR) | 0.55 | 66.35 | 67.89 | 24.18 |
| | | Word Insertion (WI) | 0.54 | 82.30 | 64.64 | 30.04 |
| | | Word Swap (WS) | 0.56 | 63.71 | 67.91 | 25.92 |
| | | Word Deletion (WD) | 0.52 | 78.23 | 64.20 | 30.57 |
| | | Insert Punctuation (IP) | 0.12 | 7.78 | 54.25 | 0.00 |
| | Sentence-level | Formal | 0.03 | 4.88 | 54.83 | 0.00 |
| | | Casual | 0.02 | 4.20 | 53.68 | 0.00 |
| | | Passive | 0.08 | 7.55 | 55.48 | 0.00 |
| | | Active | 0.13 | 8.72 | 55.59 | 0.00 |
| | | Back Translation | 0.74 | 69.22 | 67.96 | 20.70 |
| | | SCPN | 0.04 | 5.36 | 33.85 | 21.48 |
| | | Bart | 0.03 | 4.72 | 41.27 | 22.34 |
| | | Dipper | 0.07 | 3.58 | 39.27 | 20.19 |

| Watermark | Attack Category | Perturbation | BLEURT | ROUGE | Bit Acc | Dect Acc |
|---|---|---|---|---|---|---|
| Unigram-WM | Character-level | Keyboard | 0.31 | 44.98 | 55.12 | 1.65 |
| | | OCR | 0.37 | 51.62 | 55.08 | 1.72 |
| | | Character Insert (CI) | 0.31 | 45.10 | 54.82 | 1.67 |
| | | Character Replace (CR) | 0.31 | 45.10 | 55.06 | 1.67 |
| | | Character Swap (CS) | 0.35 | 46.68 | 55.21 | 1.78 |
| | | Character Delete (CD) | 0.34 | 45.86 | 55.31 | 1.82 |
| | Word-level | Synonym Replacement (SR) | 0.32 | 26.71 | 53.91 | 4.98 |
| | | Word Insertion (WI) | 0.33 | 50.89 | 53.60 | 5.20 |
| | | Word Swap (WS) | 0.33 | 48.00 | 54.53 | 6.61 |
| | | Word Deletion (WD) | 0.34 | 55.84 | 53.22 | 5.42 |
| | | Insert Punctuation (IP) | 0.33 | 0.48 | 55.13 | 0.00 |
| | Sentence-level | Formal | 0.10 | 0.22 | 14.90 | 0.00 |
| | | Casual | 0.09 | 0.20 | 13.21 | 0.00 |
| | | Passive | 0.30 | 0.98 | 48.80 | 0.00 |
| | | Active | 0.31 | 0.57 | 48.71 | 0.00 |
| | | Back Translation | 0.35 | 36.25 | 47.92 | 7.05 |
| | | SCPN | 0.03 | 0.77 | 46.37 | 1.41 |
| | | Bart | 0.06 | 0.84 | 48.13 | 3.32 |
| | | Dipper | 0.01 | 0.37 | 50.88 | 2.53 |

Table 14: Fuyu-8B text watermarks under text perturbations.

| Watermark | Attack Category | Perturbation | BLEURT | ROUGE | Bit Acc | Dect Acc |
|---|---|---|---|---|---|---|
| KGW-WM | Character-level | Keyboard | 0.38 | 65.71 | 93.15 | 36.38 |
| | | OCR | 0.53 | 79.63 | 95.93 | 34.20 |
| | | Character Insert (CI) | 0.39 | 65.90 | 68.77 | 35.20 |
| | | Character Replace (CR) | 0.38 | 65.87 | 93.84 | 35.56 |
| | | Character Swap (CS) | 0.49 | 69.28 | 92.96 | 36.06 |
| | | Character Delete (CD) | 0.45 | 65.97 | 69.68 | 36.30 |
| | Word-level | Synonym Replacement (SR) | 0.52 | 62.00 | 71.67 | 34.72 |
| | | Word Insertion (WI) | 0.53 | 75.65 | 67.62 | 37.74 |
| | | Word Swap (WS) | 0.57 | 62.39 | 71.87 | 39.82 |
| | | Word Deletion (WD) | 0.54 | 74.70 | 69.19 | 39.40 |
| | | Insert Punctuation (IP) | 0.05 | 0.00 | 53.86 | 50.54 |
| | Sentence-level | Formal | 0.05 | 0.02 | 53.30 | 48.00 |
| | | Casual | 0.03 | 2.36 | 50.20 | 32.90 |
| | | Passive | 0.02 | 8.67 | 55.03 | 40.20 |
| | | Active | 0.03 | 0.21 | 55.32 | 41.10 |
| | | Back Translation | 0.69 | 68.60 | 70.88 | 29.30 |
| | | SCPN | 0.06 | 0.00 | 50.08 | 28.53 |
| | | Bart | 0.03 | 0.06 | 50.31 | 29.90 |
| | | Dipper | 0.00 | 0.33 | 55.79 | 23.35 |

| Watermark | Attack Category | Perturbation | BLEURT | ROUGE | Bit Acc | Dect Acc |
|---|---|---|---|---|---|---|
| KTH-WM | Character-level | Keyboard | 0.38 | 72.63 | 85.55 | 37.58 |
| | | OCR | 0.52 | 82.97 | 87.04 | 36.26 |
| | | Character Insert (CI) | 0.39 | 72.74 | 67.63 | 37.76 |
| | | Character Replace (CR) | 0.38 | 72.74 | 85.85 | 37.70 |
| | | Character Swap (CS) | 0.48 | 75.23 | 85.89 | 38.88 |
| | | Character Delete (CD) | 0.45 | 72.84 | 68.19 | 36.64 |
| | Word-level | Synonym Replacement (SR) | 0.44 | 57.01 | 68.88 | 27.70 |
| | | Word Insertion (WI) | 0.46 | 72.00 | 65.96 | 43.46 |
| | | Word Swap (WS) | 0.49 | 58.95 | 69.11 | 31.66 |
| | | Word Deletion (WD) | 0.46 | 71.44 | 66.08 | 32.18 |
| | | Insert Punctuation (IP) | 0.08 | 0.02 | 53.65 | 36.22 |
| | Sentence-level | Formal | 0.08 | 0.02 | 54.96 | 33.60 |
| | | Casual | 0.08 | 0.28 | 54.94 | 32.50 |
| | | Passive | 0.06 | 2.67 | 54.82 | 33.20 |
| | | Active | 0.07 | 0.51 | 54.83 | 33.10 |
| | | Back Translation | 0.63 | 67.48 | 68.33 | 29.60 |
| | | SCPN | 0.03 | 2.84 | 54.89 | 28.44 |
| | | Bart | 0.07 | 2.18 | 60.03 | 27.45 |
| | | Dipper | 0.04 | 3.96 | 53.83 | 22.63 |

| Watermark | Attack Category | Perturbation | BLEURT | ROUGE | Bit Acc | Dect Acc |
|---|---|---|---|---|---|---|
| Blackbox-WM | Character-level | Keyboard | 0.39 | 61.86 | 82.67 | 49.04 |
| | | OCR | 0.53 | 74.36 | 84.14 | 58.80 |
| | | Character Insert (CI) | 0.40 | 61.96 | 69.04 | 47.20 |
| | | Character Replace (CR) | 0.39 | 61.98 | 82.58 | 47.68 |
| | | Character Swap (CS) | 0.49 | 65.19 | 83.16 | 52.24 |
| | | Character Delete (CD) | 0.45 | 62.07 | 69.88 | 48.72 |
| | Word-level | Synonym Replacement (SR) | 0.52 | 64.62 | 72.19 | 45.52 |
| | | Word Insertion (WI) | 0.53 | 79.48 | 68.94 | 61.32 |
| | | Word Swap (WS) | 0.58 | 64.81 | 72.34 | 38.62 |
| | | Word Deletion (WD) | 0.53 | 77.35 | 69.72 | 53.86 |
| | | Insert Punctuation (IP) | 0.04 | 0.25 | 54.28 | 0.00 |
| | Sentence-level | Formal | 0.04 | 0.01 | 55.96 | 0.00 |
| | | Casual | 0.03 | 2.33 | 55.68 | 0.00 |
| | | Passive | 0.02 | 8.23 | 55.58 | 0.00 |
| | | Active | 0.02 | 0.12 | 55.87 | 0.00 |
| | | Back Translation | 0.67 | 64.88 | 68.75 | 27.70 |
| | | SCPN | 0.01 | 3.05 | 49.21 | 0.00 |
| | | Bart | 0.00 | 4.66 | 50.78 | 0.00 |
| | | Dipper | 0.04 | 2.89 | 55.49 | 0.00 |

| Watermark | Attack Category | Perturbation | BLEURT | ROUGE | Bit Acc | Dect Acc |
|---|---|---|---|---|---|---|
| Unigram-WM | Character-level | Keyboard | 0.28 | 40.48 | 65.05 | 2.21 |
| | | OCR | 0.42 | 53.15 | 67.04 | 1.10 |
| | | Character Insert (CI) | 0.30 | 40.56 | 49.84 | 0.00 |
| | | Character Replace (CR) | 0.28 | 40.58 | 65.04 | 0.00 |
| | | Character Swap (CS) | 0.37 | 43.63 | 64.33 | 0.00 |
| | | Character Delete (CD) | 0.33 | 40.62 | 50.75 | 0.00 |
| | Word-level | Synonym Replacement (SR) | 0.40 | 46.96 | 53.28 | 4.32 |
| | | Word Insertion (WI) | 0.41 | 57.98 | 49.88 | 0.00 |
| | | Word Swap (WS) | 0.49 | 49.13 | 55.10 | 0.00 |
| | | Word Deletion (WD) | 0.43 | 58.53 | 53.43 | 0.00 |
| | | Insert Punctuation (IP) | 0.02 | 1.94 | 36.91 | 2.86 |
| | Sentence-level | Formal | 0.02 | 0.54 | 39.36 | 0.00 |
| | | Casual | 0.02 | 3.06 | 39.65 | 0.00 |
| | | Passive | 0.02 | 3.75 | 39.85 | 1.52 |
| | | Active | 0.02 | 2.93 | 39.71 | 0.00 |
| | | Back Translation | 0.51 | 44.35 | 54.19 | 0.00 |
| | | SCPN | 0.01 | 3.28 | 38.48 | 1.49 |
| | | Bart | 0.00 | 1.13 | 33.04 | 0.00 |
| | | Dipper | 0.15 | 2.07 | 35.78 | 1.13 |

Table 15: InternLM-X Composer text watermarks under text perturbations.

| Watermark | Attack Category | Perturbation | BLEURT | ROUGE | Bit Acc | Dect Acc |
|---|---|---|---|---|---|---|
| KGW-WM | Character-level | Keyboard | 0.42 | 64.78 | 87.40 | 37.38 |
| | | OCR | 0.59 | 81.32 | 90.33 | 1.42 |
| | | Character Insert (CI) | 0.44 | 65.00 | 68.48 | 37.20 |
| | | Character Replace (CR) | 0.42 | 65.01 | 87.67 | 36.02 |
| | | Character Swap (CS) | 0.51 | 69.47 | 86.48 | 16.52 |
| | | Character Delete (CD) | 0.44 | 65.00 | 69.07 | 36.08 |
| | Word-level | Synonym Replacement (SR) | 0.44 | 51.76 | 63.31 | 2.88 |
| | | Word Insertion (WI) | 0.48 | 60.34 | 61.11 | 1.76 |
| | | Word Swap (WS) | 0.57 | 47.94 | 63.61 | 15.94 |
| | | Word Deletion (WD) | 0.48 | 56.62 | 61.70 | 20.86 |
| | | Insert Punctuation (IP) | 0.09 | 0.00 | 53.02 | 6.68 |
| | Sentence-level | Formal | 0.09 | 0.00 | 54.90 | 3.70 |
| | | Casual | 0.08 | 0.00 | 54.99 | 6.20 |
| | | Passive | 0.07 | 0.00 | 54.60 | 10.80 |
| | | Active | 0.08 | 0.00 | 54.73 | 0.00 |
| | | Back Translation | 0.52 | 38.10 | 65.04 | 0.00 |
| | | SCPN | 0.04 | 1.05 | 50.17 | 20.16 |
| | | Bart | 0.06 | 0.00 | 52.24 | 15.86 |
| | | Dipper | 0.05 | 0.00 | 50.00 | 14.95 |

| Watermark | Attack Category | Perturbation | BLEURT | ROUGE | Bit Acc | Dect Acc |
|---|---|---|---|---|---|---|
| KTH-WM | Character-level | Keyboard | 0.40 | 69.21 | 89.42 | 39.54 |
| | | OCR | 0.49 | 77.41 | 85.36 | 40.66 |
| | | Character Insert (CI) | 0.35 | 71.69 | 80.02 | 42.85 |
| | | Character Replace (CR) | 0.34 | 70.78 | 77.80 | 37.42 |
| | | Character Swap (CS) | 0.33 | 68.33 | 62.53 | 29.55 |
| | | Character Delete (CD) | 0.36 | 80.41 | 23.31 | 31.86 |
| | Word-level | Synonym Replacement (SR) | 0.41 | 76.90 | 84.36 | 37.64 |
| | | Word Insertion (WI) | 0.47 | 41.31 | 85.74 | 40.89 |
| | | Word Swap (WS) | 0.55 | 50.78 | 86.93 | 42.93 |
| | | Word Deletion (WD) | 0.31 | 55.72 | 75.43 | 48.80 |
| | | Insert Punctuation (IP) | 0.02 | 1.51 | 49.21 | 34.04 |
| | Sentence-level | Formal | 0.03 | 0.03 | 45.67 | 32.28 |
| | | Casual | 0.09 | 0.00 | 44.04 | 29.37 |
| | | Passive | 0.02 | 0.01 | 36.96 | 28.63 |
| | | Active | 0.00 | 0.07 | 50.39 | 30.68 |
| | | Back Translation | 0.21 | 55.81 | 66.77 | 19.76 |
| | | SCPN | 0.00 | 2.64 | 0.00 | 20.02 |
| | | Bart | 0.02 | 1.85 | 21.56 | 20.06 |
| | | Dipper | 0.01 | 0.03 | 33.76 | 19.94 |

| Watermark | Attack Category | Perturbation | BLEURT | ROUGE | Bit Acc | Dect Acc |
|---|---|---|---|---|---|---|
| Blackbox-WM | Character-level | Keyboard | 0.38 | 52.26 | 65.34 | 26.35 |
| | | OCR | 0.53 | 62.64 | 66.43 | 21.90 |
| | | Character Insert (CI) | 0.41 | 52.30 | 54.57 | 26.97 |
| | | Character Replace (CR) | 0.33 | 52.34 | 65.21 | 27.72 |
| | | Character Swap (CS) | 0.52 | 55.08 | 65.68 | 26.41 |
| | | Character Delete (CD) | 0.48 | 52.41 | 55.23 | 25.21 |
| | Word-level | Synonym Replacement (SR) | 0.21 | 54.67 | 56.89 | 24.09 |
| | | Word Insertion (WI) | 0.55 | 68.21 | 55.02 | 21.36 |
| | | Word Swap (WS) | 0.55 | 54.49 | 56.96 | 24.33 |
| | | Word Deletion (WD) | 0.51 | 65.18 | 55.05 | 22.02 |
| | | Insert Punctuation (IP) | 0.08 | 0.21 | 42.78 | 33.36 |
| | Sentence-level | Formal | 0.07 | 0.01 | 44.19 | 33.14 |
| | | Casual | 0.05 | 1.97 | 43.97 | 28.23 |
| | | Passive | 0.01 | 6.95 | 43.89 | 21.45 |
| | | Active | 0.00 | 0.10 | 44.12 | 22.80 |
| | | Back Translation | 0.52 | 54.77 | 54.29 | 32.38 |
| | | SCPN | 0.05 | 0.16 | 42.87 | 20.31 |
| | | Bart | 0.00 | 0.21 | 44.68 | 21.74 |
| | | Dipper | 0.07 | 1.41 | 45.03 | 22.83 |

| Watermark | Attack Category | Perturbation | BLEURT | ROUGE | Bit Acc | Dect Acc |
|---|---|---|---|---|---|---|
| Unigram-WM | Character-level | Keyboard | 0.29 | 55.21 | 70.93 | 0.00 |
| | | OCR | 0.40 | 44.99 | 68.87 | 0.00 |
| | | Character Insert (CI) | 0.22 | 40.84 | 69.21 | 0.00 |
| | | Character Replace (CR) | 0.21 | 39.90 | 60.30 | 0.00 |
| | | Character Swap (CS) | 0.30 | 39.36 | 59.26 | 0.00 |
| | | Character Delete (CD) | 0.41 | 37.12 | 55.21 | 2.21 |
| | Word-level | Synonym Replacement (SR) | 0.39 | 38.00 | 70.64 | 1.10 |
| | | Word Insertion (WI) | 0.32 | 42.11 | 68.27 | 2.12 |
| | | Word Swap (WS) | 0.51 | 43.51 | 39.21 | 0.62 |
| | | Word Deletion (WD) | 0.21 | 44.98 | 39.84 | 0.00 |
| | | Insert Punctuation (IP) | 0.00 | 0.21 | 36.85 | 0.00 |
| | Sentence-level | Formal | 0.00 | 1.12 | 30.02 | 0.00 |
| | | Casual | 0.01 | 1.63 | 30.58 | 0.00 |
| | | Passive | 0.03 | 2.75 | 29.21 | 0.00 |
| | | Active | 0.00 | 3.88 | 30.64 | 0.00 |
| | | Back Translation | 0.31 | 41.86 | 44.67 | 0.00 |
| | | SCPN | 0.00 | 1.21 | 30.06 | 0.00 |
| | | Bart | 0.00 | 1.77 | 31.41 | 0.00 |
| | | Dipper | 0.00 | 1.65 | 36.84 | 0.00 |

Table 16: Instruct-BLIP text watermarks under text perturbations.

| Watermark | Attack Category | Perturbation | BLEURT | ROUGE | Bit Acc | Dect Acc |
|---|---|---|---|---|---|---|
| KGW-WM | Character-level | Keyboard | 0.44 | 0.00 | 88.15 | 42.06 |
| | | OCR | 0.52 | 0.00 | 88.16 | 57.58 |
| | | Character Insert (CI) | 0.45 | 0.00 | 79.20 | 22.44 |
| | | Character Replace (CR) | 0.41 | 0.00 | 85.64 | 25.84 |
| | | Character Swap (CS) | 0.81 | 58.32 | 98.74 | 58.32 |
| | | Character Delete (CD) | 0.77 | 0.00 | 83.59 | 68.40 |
| | Word-level | Synonym Replacement (SR) | 0.00 | 0.00 | 0.00 | 0.00 |
| | | Word Insertion (WI) | 0.20 | 0.00 | 55.26 | 0.00 |
| | | Word Swap (WS) | 0.00 | 0.00 | 0.00 | 0.00 |
| | | Word Deletion (WD) | 0.00 | 0.00 | 0.00 | 0.00 |
| | | Insert Punctuation (IP) | 0.09 | 0.00 | 40.86 | 33.08 |
| | Sentence-level | Formal | 0.36 | 0.00 | 52.58 | 88.60 |
| | | Casual | 0.31 | 0.00 | 52.32 | 52.00 |
| | | Passive | 0.37 | 0.00 | 52.63 | 0.00 |
| | | Active | 0.37 | 0.00 | 52.63 | 0.00 |
| | | Back Translation | 0.00 | 0.00 | 0.00 | 0.00 |
| | | SCPN | 0.37 | 0.00 | 52.63 | 0.00 |
| | | Bart | 0.00 | 0.00 | 41.47 | 0.00 |
| | | Dipper | 0.17 | 0.00 | 50.00 | 0.00 |

| Watermark | Attack Category | Perturbation | BLEURT | ROUGE | Bit Acc | Dect Acc |
|---|---|---|---|---|---|---|
| KTH-WM | Character-level | Keyboard | 0.46 | 21.19 | 84.59 | 39.20 |
| | | OCR | 0.57 | 25.72 | 85.93 | 46.38 |
| | | Character Insert (CI) | 0.47 | 21.21 | 73.82 | 29.44 |
| | | Character Replace (CR) | 0.44 | 21.23 | 83.18 | 30.26 |
| | | Character Swap (CS) | 0.70 | 29.55 | 89.76 | 42.42 |
| | | Character Delete (CD) | 0.66 | 21.32 | 77.36 | 40.98 |
| | Word-level | Synonym Replacement (SR) | 0.14 | 9.49 | 26.41 | 13.46 |
| | | Word Insertion (WI) | 0.22 | 11.17 | 55.82 | 13.32 |
| | | Word Swap (WS) | 0.16 | 9.52 | 26.12 | 14.66 |
| | | Word Deletion (WD) | 0.14 | 10.85 | 23.56 | 10.04 |
| | | Insert Punctuation (IP) | 0.10 | 0.91 | 46.20 | 50.56 |
| | Sentence-level | Formal | 0.22 | 1.35 | 53.55 | 53.60 |
| | | Casual | 0.21 | 1.00 | 53.45 | 48.40 |
| | | Passive | 0.22 | 1.62 | 53.51 | 57.00 |
| | | Active | 0.23 | 1.39 | 53.56 | 57.80 |
| | | Back Translation | 0.82 | 32.82 | 87.31 | 60.10 |
| | | SCPN | 0.37 | 21.38 | 30.68 | 10.66 |
| | | Bart | 0.25 | 1.66 | 31.73 | 14.95 |
| | | Dipper | 0.46 | 32.79 | 34.00 | 12.74 |

| Watermark | Attack Category | Perturbation | BLEURT | ROUGE | Bit Acc | Dect Acc |
|---|---|---|---|---|---|---|
| Blackbox-WM | Character-level | Keyboard | 0.47 | 50.46 | 90.56 | 45.86 |
| | | OCR | 0.63 | 55.15 | 81.44 | 45.76 |
| | | Character Insert (CI) | 0.49 | 49.56 | 83.63 | 45.77 |
| | | Character Replace (CR) | 0.45 | 56.78 | 85.85 | 44.89 |
| | | Character Swap (CS) | 0.58 | 62.32 | 77.42 | 31.86 |
| | | Character Delete (CD) | 0.51 | 64.12 | 77.32 | 32.75 |
| | Word-level | Synonym Replacement (SR) | 0.60 | 64.99 | 74.36 | 28.64 |
| | | Word Insertion (WI) | 0.57 | 66.46 | 64.58 | 30.32 |
| | | Word Swap (WS) | 0.62 | 62.31 | 32.18 | 39.42 |
| | | Word Deletion (WD) | 0.59 | 62.56 | 44.73 | 37.33 |
| | | Insert Punctuation (IP) | 0.15 | 2.12 | 39.47 | 0.00 |
| | Sentence-level | Formal | 0.05 | 0.00 | 36.64 | 0.00 |
| | | Casual | 0.06 | 0.00 | 37.33 | 0.00 |
| | | Passive | 0.11 | 0.00 | 30.25 | 0.00 |
| | | Active | 0.18 | 0.00 | 29.36 | 1.11 |
| | | Back Translation | 0.78 | 77.91 | 53.81 | 41.55 |
| | | SCPN | 0.04 | 0.00 | 21.56 | 0.00 |
| | | Bart | 0.00 | 0.00 | 22.61 | 1.24 |
| | | Dipper | 0.08 | 0.00 | 28.49 | 4.88 |

| Watermark | Attack Category | Perturbation | BLEURT | ROUGE | Bit Acc | Dect Acc |
|---|---|---|---|---|---|---|
| Unigram-WM | Character-level | Keyboard | 0.41 | 32.67 | 85.08 | 28.03 |
| | | OCR | 0.49 | 35.33 | 85.33 | 18.74 |
| | | Character Insert (CI) | 0.39 | 32.20 | 80.92 | 21.46 |
| | | Character Replace (CR) | 0.39 | 32.08 | 84.69 | 23.46 |
| | | Character Swap (CS) | 0.54 | 51.68 | 85.71 | 24.28 |
| | | Character Delete (CD) | 0.51 | 32.39 | 82.58 | 22.48 |
| | Word-level | Synonym Replacement (SR) | 0.17 | 13.92 | 32.84 | 17.97 |
| | | Word Insertion (WI) | 0.18 | 19.70 | 33.12 | 17.76 |
| | | Word Swap (WS) | 0.17 | 17.10 | 30.92 | 15.64 |
| | | Word Deletion (WD) | 0.17 | 19.92 | 29.88 | 15.04 |
| | | Insert Punctuation (IP) | 0.17 | 1.00 | 49.66 | 21.08 |
| | Sentence-level | Formal | 0.12 | 1.36 | 31.85 | 15.68 |
| | | Casual | 0.12 | 1.43 | 31.88 | 16.58 |
| | | Passive | 0.11 | 1.49 | 31.96 | 15.58 |
| | | Active | 0.11 | 1.48 | 31.98 | 14.09 |
| | | Back Translation | 0.32 | 27.74 | 38.36 | 16.08 |
| | | SCPN | 0.03 | 1.31 | 20.59 | 10.77 |
| | | Bart | 0.18 | 1.89 | 21.66 | 11.53 |
| | | Dipper | 0.12 | 1.66 | 20.78 | 10.89 |

Table 17: LLaVA 1.6 text watermarks under text perturbations.

| Watermark | Attack Category | Perturbation | BLEURT | ROUGE | Bit Acc | Dect Acc |
|---|---|---|---|---|---|---|
| KGW-WM | Character-level | Keyboard | 0.45 | 68.64 | 93.56 | 56.34 |
| | | OCR | 0.58 | 79.10 | 96.48 | 91.60 |
| | | Character Insert (CI) | 0.47 | 68.75 | 67.49 | 61.82 |
| | | Character Replace (CR) | 0.45 | 68.75 | 94.45 | 58.26 |
| | | Character Swap (CS) | 0.54 | 72.15 | 93.97 | 49.66 |
| | | Character Delete (CD) | 0.53 | 68.75 | 68.11 | 64.62 |
| | Word-level | Synonym Replacement (SR) | 0.61 | 64.58 | 71.94 | 39.74 |
| | | Word Insertion (WI) | 0.56 | 75.15 | 69.27 | 91.44 |
| | | Word Swap (WS) | 0.61 | 58.91 | 74.68 | 53.88 |
| | | Word Deletion (WD) | 0.56 | 71.80 | 70.69 | 60.94 |
| | | Insert Punctuation (IP) | 0.06 | 0.00 | 53.32 | 92.18 |
| | Sentence-level | Formal | 0.05 | 0.22 | 54.49 | 73.60 |
| | | Casual | 0.07 | 1.81 | 54.76 | 58.70 |
| | | Passive | 0.04 | 7.07 | 54.67 | 99.50 |
| | | Active | 0.06 | 0.15 | 55.01 | 0.00 |
| | | Back Translation | 0.83 | 90.32 | 67.23 | 0.00 |
| | | SCPN | 0.00 | 2.51 | 48.31 | 77.88 |
| | | Bart | 0.08 | 1.38 | 44.86 | 53.64 |
| | | Dipper | 0.03 | 4.51 | 60.75 | 72.33 |

| Watermark | Attack Category | Perturbation | BLEURT | ROUGE | Bit Acc | Dect Acc |
|---|---|---|---|---|---|---|
| KTH-WM | Character-level | Keyboard | 0.44 | 59.05 | 81.25 | 33.06 |
| | | OCR | 0.56 | 70.54 | 82.74 | 34.70 |
| | | Character Insert (CI) | 0.46 | 59.03 | 66.30 | 35.10 |
| | | Character Replace (CR) | 0.43 | 59.09 | 80.74 | 33.36 |
| | | Character Swap (CS) | 0.55 | 62.00 | 81.44 | 33.74 |
| | | Character Delete (CD) | 0.49 | 59.14 | 67.24 | 33.18 |
| | Word-level | Synonym Replacement (SR) | 0.44 | 37.64 | 67.30 | 32.80 |
| | | Word Insertion (WI) | 0.45 | 44.25 | 65.47 | 34.92 |
| | | Word Swap (WS) | 0.47 | 37.09 | 66.82 | 35.62 |
| | | Word Deletion (WD) | 0.43 | 42.82 | 64.81 | 35.16 |
| | | Insert Punctuation (IP) | 0.08 | 0.50 | 53.94 | 37.02 |
| | Sentence-level | Formal | 0.09 | 0.57 | 55.63 | 35.40 |
| | | Casual | 0.08 | 0.49 | 55.61 | 38.00 |
| | | Passive | 0.07 | 2.05 | 55.31 | 35.10 |
| | | Active | 0.08 | 0.90 | 55.50 | 37.90 |
| | | Back Translation | 0.68 | 66.43 | 72.95 | 36.50 |
| | | SCPN | 0.09 | 1.21 | 54.67 | 39.00 |
| | | Bart | 0.00 | 0.00 | 50.31 | 39.45 |
| | | Dipper | 0.09 | 0.31 | 55.10 | 39.30 |

| Watermark | Attack Category | Perturbation | BLEURT | ROUGE | Bit Acc | Dect Acc |
|---|---|---|---|---|---|---|
| Blackbox-WM | Character-level | Keyboard | 0.41 | 65.80 | 81.00 | 41.70 |
| | | OCR | 0.60 | 79.50 | 83.00 | 51.05 |
| | | Character Insert (CI) | 0.45 | 66.20 | 68.50 | 40.55 |
| | | Character Replace (CR) | 0.43 | 69.13 | 81.50 | 41.25 |
| | | Character Swap (CS) | 0.56 | 66.10 | 81.75 | 42.62 |
| | | Character Delete (CD) | 0.51 | 66.40 | 69.50 | 40.02 |
| | Word-level | Synonym Replacement (SR) | 0.58 | 82.50 | 67.00 | 24.33 |
| | | Word Insertion (WI) | 0.57 | 63.90 | 64.00 | 30.22 |
| | | Word Swap (WS) | 0.59 | 78.30 | 68.00 | 26.03 |
| | | Word Deletion (WD) | 0.51 | 58.77 | 64.50 | 30.85 |
| | | Insert Punctuation (IP) | 0.14 | 7.90 | 54.00 | 0.00 |
| | Sentence-level | Formal | 0.05 | 4.90 | 55.00 | 0.00 |
| | | Casual | 0.04 | 4.30 | 54.50 | 0.00 |
| | | Passive | 0.10 | 7.60 | 56.12 | 0.00 |
| | | Active | 0.11 | 8.80 | 68.94 | 0.00 |
| | | Back Translation | 0.16 | 69.40 | 47.31 | 21.03 |
| | | SCPN | 0.00 | 0.00 | 33.60 | 25.84 |
| | | Bart | 0.00 | 1.84 | 17.58 | 10.44 |
| | | Dipper | 0.01 | 0.83 | 35.62 | 5.82 |

| Watermark | Attack Category | Perturbation | BLEURT | ROUGE | Bit Acc | Dect Acc |
|---|---|---|---|---|---|---|
| Unigram-WM | Character-level | Keyboard | 0.48 | 80.42 | 71.23 | 3.29 |
| | | OCR | 0.57 | 84.97 | 71.46 | 4.95 |
| | | Character Insert (CI) | 0.47 | 80.40 | 67.08 | 3.63 |
| | | Character Replace (CR) | 0.47 | 80.36 | 70.89 | 5.14 |
| | | Character Swap (CS) | 0.56 | 81.57 | 71.42 | 3.31 |
| | | Character Delete (CD) | 0.54 | 80.54 | 67.65 | 4.27 |
| | Word-level | Synonym Replacement (SR) | 0.34 | 37.30 | 65.12 | 4.21 |
| | | Word Insertion (WI) | 0.35 | 55.48 | 63.27 | 9.34 |
| | | Word Swap (WS) | 0.35 | 47.33 | 65.30 | 2.97 |
| | | Word Deletion (WD) | 0.36 | 56.26 | 62.71 | 6.12 |
| | | Insert Punctuation (IP) | 0.21 | 0.44 | 53.44 | 4.29 |
| | Sentence-level | Formal | 0.21 | 0.47 | 54.76 | 7.46 |
| | | Casual | 0.21 | 0.54 | 54.77 | 7.86 |
| | | Passive | 0.21 | 0.95 | 54.58 | 2.67 |
| | | Active | 0.22 | 0.72 | 54.59 | 3.07 |
| | | Back Translation | 0.57 | 66.62 | 64.67 | 8.36 |
| | | SCPN | 0.48 | 1.37 | 33.75 | 3.74 |
| | | Bart | 0.31 | 2.63 | 27.93 | 1.37 |
| | | Dipper | 0.17 | 1.84 | 30.41 | 2.91 |

Table 18: MiniGPT-4 text watermarks under text perturbations.

| Watermark | Attack Category | Perturbation | BLEURT | ROUGE | Bit Acc | Dect Acc |
|---|---|---|---|---|---|---|
| KGW-WM | Character-level | Keyboard | 0.64 | 85.94 | 70.63 | 22.68 |
| | | OCR | 0.47 | 64.38 | 65.31 | 19.69 |
| | | Character Insert (CI) | 0.63 | 83.52 | 70.57 | 23.40 |
| | | Character Replace (CR) | 0.82 | 65.23 | 76.50 | 27.54 |
| | | Character Swap (CS) | 0.39 | 51.51 | 62.83 | 19.27 |
| | | Character Delete (CD) | 0.36 | 48.14 | 62.24 | 19.28 |
| | Word-level | Synonym Replacement (SR) | 0.77 | 96.73 | 75.15 | 27.83 |
| | | Word Insertion (WI) | 0.56 | 70.10 | 68.52 | 24.01 |
| | | Word Swap (WS) | 0.25 | 31.28 | 58.72 | 18.19 |
| | | Word Deletion (WD) | 0.47 | 56.91 | 65.66 | 22.97 |
| | | Insert Punctuation (IP) | 0.21 | 24.71 | 57.58 | 18.24 |
| | Sentence-level | Formal | 0.19 | 21.28 | 56.97 | 18.24 |
| | | Casual | 0.34 | 38.18 | 61.65 | 21.58 |
| | | Passive | 0.19 | 26.94 | 45.01 | 11.44 |
| | | Active | 0.17 | 24.92 | 45.83 | 12.33 |
| | | Back Translation | 0.09 | 5.02 | 53.89 | 17.82 |
| | | SCPN | 0.19 | 3.88 | 33.51 | 10.86 |
| | | Bart | 0.15 | 10.63 | 32.85 | 11.88 |
| | | Dipper | 0.17 | 11.58 | 35.57 | 12.75 |

| Watermark | Attack Category | Perturbation | BLEURT | ROUGE | Bit Acc | Dect Acc |
|---|---|---|---|---|---|---|
| KTH-WM | Character-level | Keyboard | 0.57 | 80.81 | 76.41 | 24.59 |
| | | OCR | 0.44 | 68.70 | 64.65 | 20.53 |
| | | Character Insert (CI) | 0.56 | 88.03 | 75.94 | 25.64 |
| | | Character Replace (CR) | 0.70 | 88.03 | 88.69 | 31.32 |
| | | Character Swap (CS) | 0.38 | 53.39 | 58.72 | 20.03 |
| | | Character Delete (CD) | 0.36 | 49.24 | 57.26 | 20.07 |
| | Word-level | Synonym Replacement (SR) | 0.66 | 99.27 | 85.23 | 31.81 |
| | | Word Insertion (WI) | 0.50 | 70.87 | 70.60 | 26.61 |
| | | Word Swap (WS) | 0.27 | 29.77 | 49.07 | 18.68 |
| | | Word Deletion (WD) | 0.43 | 55.86 | 64.03 | 25.24 |
| | | Insert Punctuation (IP) | 0.24 | 21.66 | 46.26 | 18.80 |
| | Sentence-level | Formal | 0.22 | 17.44 | 44.76 | 18.82 |
| | | Casual | 0.33 | 34.44 | 54.79 | 23.43 |
| | | Passive | 0.06 | 34.09 | 18.36 | 9.57 |
| | | Active | 0.05 | 32.62 | 19.96 | 10.82 |
| | | Back Translation | 0.14 | 2.03 | 37.37 | 18.36 |
| | | SCPN | 0.07 | 3.79 | 18.77 | 21.99 |
| | | Bart | 0.08 | 4.88 | 17.38 | 22.54 |
| | | Dipper | 0.00 | 3.72 | 13.75 | 19.58 |

| Watermark | Attack Category | Perturbation | BLEURT | ROUGE | Bit Acc | Dect Acc |
|---|---|---|---|---|---|---|
| Blackbox-WM | Character-level | Keyboard | 0.65 | 95.92 | 85.39 | 58.32 |
| | | OCR | 0.49 | 72.82 | 77.19 | 43.55 |
| | | Character Insert (CI) | 0.64 | 92.24 | 83.34 | 54.34 |
| | | Character Replace (CR) | 0.82 | 74.33 | 90.40 | 66.74 |
| | | Character Swap (CS) | 0.41 | 57.66 | 70.85 | 31.78 |
| | | Character Delete (CD) | 0.39 | 53.56 | 69.07 | 28.43 |
| | Word-level | Synonym Replacement (SR) | 0.77 | 73.71 | 85.60 | 57.70 |
| | | Word Insertion (WI) | 0.57 | 75.32 | 75.61 | 39.74 |
| | | Word Swap (WS) | 0.27 | 34.22 | 61.32 | 14.14 |
| | | Word Deletion (WD) | 0.48 | 60.40 | 69.76 | 29.00 |
| | | Insert Punctuation (IP) | 0.23 | 26.20 | 57.81 | 7.55 |
| | Sentence-level | Formal | 0.21 | 22.03 | 56.00 | 4.16 |
| | | Casual | 0.35 | 39.10 | 61.36 | 0.00 |
| | | Passive | 0.15 | 29.48 | 37.80 | 0.00 |
| | | Active | 0.13 | 27.94 | 37.91 | 0.00 |
| | | Back Translation | 0.10 | 2.74 | 47.87 | 10.97 |
| | | SCPN | 0.15 | 3.83 | 50.31 | 0.00 |
| | | Bart | 0.13 | 3.94 | 52.73 | 1.41 |
| | | Dipper | 0.17 | 4.34 | 55.79 | 8.36 |

| Watermark | Attack Category | Perturbation | BLEURT | ROUGE | Bit Acc | Dect Acc |
|---|---|---|---|---|---|---|
| Unigram-WM | Character-level | Keyboard | 0.38 | 65.69 | 65.10 | 3.61 |
| | | OCR | 0.33 | 48.79 | 55.35 | 2.08 |
| | | Character Insert (CI) | 0.38 | 62.63 | 64.39 | 4.00 |
| | | Character Replace (CR) | 0.45 | 78.40 | 74.61 | 6.13 |
| | | Character Swap (CS) | 0.30 | 37.23 | 50.04 | 1.87 |
| | | Character Delete (CD) | 0.30 | 34.06 | 48.69 | 1.88 |
| | Word-level | Synonym Replacement (SR) | 0.43 | 70.12 | 71.30 | 6.30 |
| | | Word Insertion (WI) | 0.36 | 49.40 | 59.22 | 4.33 |
| | | Word Swap (WS) | 0.26 | 19.48 | 41.53 | 1.34 |
| | | Word Deletion (WD) | 0.33 | 38.21 | 53.55 | 3.81 |
| | | Insert Punctuation (IP) | 0.25 | 13.28 | 38.91 | 1.37 |
| | Sentence-level | Formal | 0.24 | 10.07 | 37.53 | 1.37 |
| | | Casual | 0.29 | 22.21 | 45.53 | 0.00 |
| | | Passive | 0.12 | 27.57 | 15.70 | 0.00 |
| | | Active | 0.12 | 26.66 | 16.84 | 0.00 |
| | | Back Translation | 0.21 | 4.68 | 30.85 | 1.18 |
| | | SCPN | 0.44 | 2.55 | 29.67 | 4.74 |
| | | Bart | 0.36 | 4.21 | 30.51 | 0.00 |
| | | Dipper | 0.45 | 5.42 | 31.87 | 3.21 |

Table 19: mPLUG-Owl2 text watermarks under text perturbations.

| Watermark | Attack Category | Perturbation | BLEURT | ROUGE | Bit Acc | Dect Acc |
|---|---|---|---|---|---|---|
| KGW-WM | Character-level | Keyboard | 0.62 | 66.29 | 67.84 | 38.48 |
| | | OCR | 0.46 | 72.11 | 85.61 | 35.48 |
| | | Character Insert (CI) | 0.61 | 92.23 | 94.93 | 39.81 |
| | | Character Replace (CR) | 0.79 | 75.14 | 65.61 | 44.60 |
| | | Character Swap (CS) | 0.37 | 55.97 | 76.37 | 35.80 |
| | | Character Delete (CD) | 0.35 | 51.59 | 73.77 | 36.07 |
| | Word-level | Synonym Replacement (SR) | 0.74 | 88.75 | 98.67 | 45.71 |
| | | Word Insertion (WI) | 0.54 | 74.05 | 83.76 | 41.79 |
| | | Word Swap (WS) | 0.24 | 31.11 | 62.41 | 35.67 |
| | | Word Deletion (WD) | 0.44 | 58.28 | 75.16 | 41.18 |
| | | Insert Punctuation (IP) | 0.10 | 22.53 | 57.31 | 36.25 |
| | Sentence-level | Formal | 0.08 | 18.08 | 54.68 | 36.51 |
| | | Casual | 0.01 | 35.76 | 62.81 | 40.44 |
| | | Passive | 0.09 | 35.82 | 27.54 | 29.58 |
| | | Active | 0.07 | 34.33 | 27.80 | 30.83 |
| | | Back Translation | 0.07 | 2.46 | 42.83 | 37.10 |
| | | SCPN | 0.03 | 3.53 | 28.64 | 33.41 |
| | | Bart | 0.00 | 0.00 | 29.99 | 38.90 |
| | | Dipper | 0.06 | 2.14 | 20.05 | 37.52 |

| Watermark | Attack Category | Perturbation | BLEURT | ROUGE | Bit Acc | Dect Acc |
|---|---|---|---|---|---|---|
| KTH-WM | Character-level | Keyboard | 0.58 | 82.91 | 89.41 | 40.02 |
| | | OCR | 0.44 | 76.98 | 79.90 | 37.17 |
| | | Character Insert (CI) | 0.57 | 67.78 | 87.10 | 39.72 |
| | | Character Replace (CR) | 0.71 | 61.52 | 95.34 | 42.61 |
| | | Character Swap (CS) | 0.36 | 58.69 | 72.64 | 35.51 |
| | | Character Delete (CD) | 0.34 | 53.64 | 70.59 | 35.07 |
| | Word-level | Synonym Replacement (SR) | 0.67 | 78.22 | 89.87 | 41.52 |
| | | Word Insertion (WI) | 0.49 | 76.48 | 78.28 | 38.01 |
| | | Word Swap (WS) | 0.24 | 30.76 | 61.69 | 32.88 |
| | | Word Deletion (WD) | 0.41 | 58.99 | 71.54 | 36.28 |
| | | Insert Punctuation (IP) | 0.21 | 20.86 | 57.67 | 32.03 |
| | Sentence-level | Formal | 0.02 | 5.74 | 55.60 | 31.59 |
| | | Casual | 0.03 | 3.96 | 61.87 | 33.84 |
| | | Passive | 0.01 | 1.96 | 34.49 | 25.23 |
| | | Active | 0.01 | 0.81 | 34.66 | 25.51 |
| | | Back Translation | 0.09 | 7.63 | 46.28 | 29.49 |
| | | SCPN | 0.00 | 2.58 | 33.89 | 22.04 |
| | | Bart | 0.01 | 3.21 | 36.53 | 27.49 |
| | | Dipper | 0.02 | 3.33 | 31.05 | 28.81 |

| Watermark | Attack Category | Perturbation | BLEURT | ROUGE | Bit Acc | Dect Acc |
|---|---|---|---|---|---|---|
| Blackbox-WM | Character-level | Keyboard | 0.63 | 52.81 | 87.64 | 73.97 |
| | | OCR | 0.46 | 69.28 | 79.20 | 55.15 |
| | | Character Insert (CI) | 0.62 | 89.23 | 85.57 | 69.76 |
| | | Character Replace (CR) | 0.80 | 61.92 | 92.88 | 86.47 |
| | | Character Swap (CS) | 0.38 | 54.03 | 72.73 | 41.25 |
| | | Character Delete (CD) | 0.35 | 49.93 | 70.91 | 37.37 |
| | Word-level | Synonym Replacement (SR) | 0.74 | 71.33 | 88.00 | 76.14 |
| | | Word Insertion (WI) | 0.54 | 72.38 | 77.71 | 53.16 |
| | | Word Swap (WS) | 0.23 | 30.42 | 62.99 | 20.18 |
| | | Word Deletion (WD) | 0.44 | 57.30 | 71.72 | 40.11 |
| | | Insert Punctuation (IP) | 0.19 | 22.40 | 59.41 | 12.56 |
| | Sentence-level | Formal | 0.07 | 18.23 | 57.57 | 8.62 |
| | | Casual | 0.01 | 5.79 | 63.12 | 1.39 |
| | | Passive | 0.00 | 0.28 | 38.83 | 3.19 |
| | | Active | 0.08 | 0.62 | 38.97 | 2.64 |
| | | Back Translation | 0.06 | 1.13 | 49.27 | 9.17 |
| | | SCPN | 0.00 | 1.31 | 30.04 | 3.53 |
| | | Bart | 0.06 | 1.22 | 30.78 | 2.55 |
| | | Dipper | 0.09 | 1.85 | 30.95 | 2.78 |

| Watermark | Attack Category | Perturbation | BLEURT | ROUGE | Bit Acc | Dect Acc |
|---|---|---|---|---|---|---|
| Unigram-WM | Character-level | Keyboard | 0.48 | 63.53 | 68.48 | 1.71 |
| | | OCR | 0.35 | 47.33 | 60.55 | 0.85 |
| | | Character Insert (CI) | 0.48 | 61.44 | 66.76 | 1.82 |
| | | Character Replace (CR) | 0.62 | 77.46 | 73.85 | 2.90 |
| | | Character Swap (CS) | 0.28 | 37.32 | 54.77 | 0.60 |
| | | Character Delete (CD) | 0.27 | 34.66 | 53.16 | 0.56 |
| | Word-level | Synonym Replacement (SR) | 0.58 | 70.69 | 69.58 | 2.84 |
| | | Word Insertion (WI) | 0.42 | 50.71 | 59.90 | 1.75 |
| | | Word Swap (WS) | 0.18 | 21.67 | 45.99 | 0.13 |
| | | Word Deletion (WD) | 0.34 | 40.60 | 54.44 | 1.38 |
| | | Insert Punctuation (IP) | 0.05 | 16.48 | 42.83 | 0.05 |
| | Sentence-level | Formal | 0.03 | 3.78 | 41.20 | 0.00 |
| | | Casual | 0.04 | 6.22 | 46.62 | 0.87 |
| | | Passive | 0.06 | 0.42 | 23.59 | 1.94 |
| | | Active | 0.05 | 0.06 | 23.85 | 1.74 |
| | | Back Translation | 0.05 | 1.09 | 33.80 | 0.29 |
| | | SCPN | 0 | 0.09 | 29.41 | 1.53 |
| | | Bart | 0.06 | 0.03 | 28.79 | 1.23 |
| | | Dipper | 0.05 | 0.01 | 26.47 | 1.88 |

Table 20: Qwen-VL text watermarks under text perturbations.

| Watermark | Attack Category | Perturbation | BLEURT | ROUGE | Bit Acc | Dect Acc |
|---|---|---|---|---|---|---|
| KGW-WM | Character-level | Keyboard | 0.64 | 65.57 | 90.76 | 30.89 |
| | | OCR | 0.49 | 51.60 | 80.30 | 15.60 |
| | | Character Insert (CI) | 0.62 | 69.54 | 88.15 | 35.16 |
| | | Character Replace (CR) | 0.76 | 10.11 | 97.14 | 56.91 |
| | | Character Swap (CS) | 0.40 | 53.03 | 72.23 | 14.10 |
| | | Character Delete (CD) | 0.37 | 47.79 | 69.96 | 14.39 |
| | Word-level | Synonym Replacement (SR) | 0.70 | 96.03 | 91.03 | 59.14 |
| | | Word Insertion (WI) | 0.52 | 66.84 | 78.30 | 39.52 |
| | | Word Swap (WS) | 0.26 | 25.11 | 60.09 | 9.47 |
| | | Word Deletion (WD) | 0.42 | 49.72 | 70.85 | 34.57 |
| | | Insert Punctuation (IP) | 0.21 | 14.80 | 55.61 | 10.19 |
| | Sentence-level | Formal | 0.02 | 0.49 | 53.31 | 10.42 |
| | | Casual | 0.03 | 0.12 | 60.15 | 28.05 |
| | | Passive | 0.01 | 0.70 | 30.11 | 24.51 |
| | | Active | 0.01 | 0.39 | 30.26 | 19.61 |
| | | Back Translation | 0.07 | 0.34 | 42.95 | 9.19 |
| | | SCPN | 0.00 | 0.54 | 40.03 | 9.99 |
| | | Bart | 0.02 | 0.59 | 44.65 | 9.12 |
| | | Dipper | 0.01 | 0.79 | 42.85 | 8.13 |
| **Watermark** | **Attack Category** | **Perturbation** | **BLEURT** | **ROUGE** | **Bit Acc** | **Dect Acc** |
| KTH-WM | Character-level | Keyboard | 0.57 | 70.00 | 92.83 | 45.41 |
| | | OCR | 0.43 | 74.63 | 77.60 | 40.10 |
| | | Character Insert (CI) | 0.55 | 94.32 | 91.23 | 44.73 |
| | | Character Replace (CR) | 0.67 | 66.84 | 66.68 | 49.99 |
| | | Character Swap (CS) | 0.33 | 55.89 | 68.65 | 36.83 |
| | | Character Delete (CD) | 0.30 | 50.66 | 66.32 | 35.96 |
| | Word-level | Synonym Replacement (SR) | 0.60 | 72.93 | 80.83 | 47.78 |
| | | Word Insertion (WI) | 0.43 | 71.95 | 82.00 | 41.23 |
| | | Word Swap (WS) | 0.18 | 27.49 | 54.54 | 31.71 |
| | | Word Deletion (WD) | 0.33 | 54.35 | 72.77 | 37.93 |
| | | Insert Punctuation (IP) | 0.12 | 17.22 | 50.00 | 30.02 |
| | Sentence-level | Formal | 0.09 | 1.91 | 47.62 | 29.14 |
| | | Casual | 0.02 | 9.12 | 59.67 | 33.22 |
| | | Passive | 0.02 | 4.46 | 13.55 | 17.27 |
| | | Active | 0.02 | 0.72 | 15.04 | 17.73 |
| | | Back Translation | 0.00 | 0.08 | 36.33 | 24.99 |
| | | SCPN | 0.00 | 0.01 | 18.68 | 18.46 |
| | | Bart | 0.03 | 0.04 | 19.68 | 19.00 |
| | | Dipper | 0.06 | 2.31 | 16.38 | 17.62 |
| **Watermark** | **Attack Category** | **Perturbation** | **BLEURT** | **ROUGE** | **Bit Acc** | **Dect Acc** |
| Blackbox-WM | Character-level | Keyboard | 0.61 | 78.45 | 69.26 | 26.62 |
| | | OCR | 0.45 | 58.54 | 62.59 | 24.32 |
| | | Character Insert (CI) | 0.59 | 75.43 | 67.62 | 27.60 |
| | | Character Replace (CR) | 0.75 | 94.64 | 73.39 | 31.23 |
| | | Character Swap (CS) | 0.36 | 45.66 | 57.47 | 24.52 |
| | | Character Delete (CD) | 0.33 | 42.20 | 56.03 | 24.71 |
| | Word-level | Synonym Replacement (SR) | 0.69 | 85.70 | 69.53 | 32.02 |
| | | Word Insertion (WI) | 0.50 | 61.21 | 61.40 | 29.03 |
| | | Word Swap (WS) | 0.21 | 25.71 | 49.76 | 24.36 |
| | | Word Deletion (WD) | 0.40 | 48.45 | 56.66 | 28.53 |
| | | Insert Punctuation (IP) | 0.17 | 8.94 | 46.93 | 24.77 |
| | Sentence-level | Formal | 0.04 | 5.41 | 45.46 | 24.95 |
| | | Casual | 0.07 | 0.03 | 49.86 | 27.92 |
| | | Passive | 0.02 | 0.01 | 30.65 | 19.65 |
| | | Active | 0.09 | 0.06 | 30.76 | 20.58 |
| | | Back Translation | 0.03 | 0.10 | 38.90 | 25.34 |
| | | SCPN | 0.00 | 0.00 | 39.68 | 18.80 |
| | | Bart | 0.00 | 0.03 | 33.29 | 17.36 |
| | | Dipper | 0.00 | 0.06 | 38.48 | 16.56 |
| **Watermark** | **Attack Category** | **Perturbation** | **BLEURT** | **ROUGE** | **Bit Acc** | **Dect Acc** |
| Unigram-WM | Character-level | Keyboard | 0.45 | 60.66 | 80.47 | 0.92 |
| | | OCR | 0.33 | 45.61 | 67.28 | 0.43 |
| | | Character Insert (CI) | 0.44 | 57.90 | 76.97 | 0.98 |
| | | Character Replace (CR) | 0.56 | 71.90 | 88.09 | 0.00 |
| | | Character Swap (CS) | 0.26 | 35.26 | 56.84 | 0.29 |
| | | Character Delete (CD) | 0.24 | 32.43 | 53.88 | 0.00 |
| | Word-level | Synonym Replacement (SR) | 0.52 | 64.48 | 80.10 | 1.58 |
| | | Word Insertion (WI) | 0.36 | 46.03 | 64.07 | 0.96 |
| | | Word Swap (WS) | 0.14 | 19.39 | 41.19 | 0.00 |
| | | Word Deletion (WD) | 0.29 | 36.03 | 54.52 | 0.00 |
| | | Insert Punctuation (IP) | 0.10 | 3.84 | 35.36 | 0.00 |
| | Sentence-level | Formal | 0.08 | 0.96 | 32.36 | 0.00 |
| | | Casual | 0.02 | 0.07 | 40.78 | 0.00 |
| | | Passive | 0.02 | 0.06 | 3.13 | 0.00 |
| | | Active | 0.02 | 0.08 | 3.19 | 0.00 |
| | | Back Translation | 0.01 | 0.02 | 18.94 | 0.21 |
| | | SCPN | 0.00 | 0.04 | 3.68 | 0.00 |
| | | Bart | 0.00 | 0.02 | 3.52 | 0.00 |
| | | Dipper | 0.07 | 0.07 | 4.42 | 0.00 |

