# OpenReview forum: "Evaluating Durability: Benchmark Insights into Image and Text Watermarking"
_DMLR — Accepted by DMLR_

### Review · Reviewer_oaTv · 2024-10-04

**Recommendation:** 4
**Confidence:** 1

**Summary Of Contributions:**

This paper studies how robust multimodal watermarks are when images or texts are perturbed in different ways. The authors test four image and four text watermark methods to see how well they work when the content is changed by different types of noise or distortion.
They found that Zoom Blur has the biggest effect on image watermarks, while Glass Blur affects them the least. For text, the Casual style of writing changes watermarks the most, while OCR errors (from converting images of text to digital text) affect them the least. Among the methods tested, SSL-WM for images and KGW-WM for text, are more stable when content changes.

**Strengths:**

1. The paper is well-written and easy to follow. The findings and takeaways are clearly demonstrated.
2. The problem of watermark robustness is important. This is an important contribution to the field and has clear real-world applications, particularly in copyright protection and content verification.
3. It provides a comprehensive evaluation for the given settings. It covers multiple models and perturbation methods, etc., providing insights into the strengths and weaknesses of the tested watermarking techniques.

**Audience:**

Yes

**Broader Impact Concerns:**

The paper has already included a broader impact statement.

**Claims And Evidence:**

Yes

**Datasets And Benchmarks:**

Yes

**Extended Submissions:**

N/A

**Limitations:**

1.  Including more types of perturbations, watermarking methods, and models would enhance the study. However, the current set is sufficient to me, and the paper appropriately claims the scope of its contribution.
2. The paper does not introduce many novel methods, but as an evaluation benchmark, this is acceptable.

**Requested Changes:**

N/A

**Strengths And Weaknesses:**

The strength and weaknesses are listed as follows.

---

### Review · Reviewer_JS8X · 2024-10-05

**Recommendation:** 3
**Confidence:** 2

**Summary Of Contributions:**

This paper introduces a comprehensive benchmark assessing the robustness of text and image watermarks against various perturbations. Specifically, it evaluates the resilience of four image watermarking methods and four text watermarking systems against 100 image perturbation techniques and 63 text perturbation methods. Additionally, the authors have open-sourced their code, which has the potential to facilitate further research and advancements in watermarking strategies.

**Strengths:**

- The paper investigates the critical property of watermark robustness, laying a strong foundation for future research by providing a benchmark that quantitatively evaluates this property.

- The evaluation methods are representative and new, and the range of perturbations considered is extensive. The comprehensive experiments yield valuable insights into the behavior of different watermarking techniques.

- The paper is well-structured and clearly written.

**Audience:**

Yes

**Broader Impact Concerns:**

I see no ethical concerns involved.

**Claims And Evidence:**

Yes

**Datasets And Benchmarks:**

Yes, it has provided sufficient details to support reproducibility.

**Extended Submissions:**

N/A

**Limitations:**

- The role of models in the watermarking process remains unclear. The paper predominantly focuses on data watermarking, with most methods appearing to be model-independent, as indicated by the descriptions. However, the specific role of models in the watermark generation and detection process is not clearly explained. For example, Table 2 presents results for various watermarking methods and models, but it is not evident what the models represent or their contribution to watermarking.

- Although the paper claims to benchmark multi-modal watermarking, the evaluation only covers image and text watermarks. Similarly, the perturbation tests are restricted to image and text perturbations. It is unclear why the term "multi-modal" is applied, given this limited scope.

- The results primarily focus on detection accuracy and the drop in detection accuracy after perturbations. It would be beneficial to also include metrics on the original image and text quality to provide a clearer understanding of how different perturbation techniques affect the quality of the content.

- One crucial aspect of watermarking is the trade-off between detection accuracy and content quality. If an adversary injects substantial noise to disrupt the watermark, the underlying content may also be degraded, undermining the value of invisible watermarking. It would strengthen the paper to include an analysis of this trade-off, perhaps through curves illustrating detection accuracy versus content quality as perturbation levels increase. Additionally, discussing which watermarking methods offer the best trade-off, and identifying which perturbations most effectively disrupt this trade-off, would provide deeper insights.

**Requested Changes:**

- Clarify the role of models in the watermark generation and detection processes.
- Explain the focus on multi-modal watermarking and how it aligns with the evaluations.
- Include metrics on the original image and text quality as part of the evaluation.
- Explore the trade-off between detection accuracy and content quality.

**Strengths And Weaknesses:**

Strengths:

- The paper investigates the critical property of watermark robustness, laying a strong foundation for future research by providing a benchmark that quantitatively evaluates this property.

- The evaluation methods are representative and new, and the range of perturbations considered is extensive. The comprehensive experiments yield valuable insights into the behavior of different watermarking techniques.

- The paper is well-structured and clearly written.

Weaknesses:

- The role of models in the watermarking process remains unclear. The paper predominantly focuses on data watermarking, with most methods appearing to be model-independent, as indicated by the descriptions. However, the specific role of models in the watermark generation and detection process is not clearly explained. For example, Table 2 presents results for various watermarking methods and models, but it is not evident what the models represent or their contribution to watermarking.

- Although the paper claims to benchmark multi-modal watermarking, the evaluation only covers image and text watermarks. Similarly, the perturbation tests are restricted to image and text perturbations. It is unclear why the term "multi-modal" is applied, given this limited scope.

- The results primarily focus on detection accuracy and the drop in detection accuracy after perturbations. It would be beneficial to also include metrics on the original image and text quality to provide a clearer understanding of how different perturbation techniques affect the quality of the content.

- One crucial aspect of watermarking is the trade-off between detection accuracy and content quality. If an adversary injects substantial noise to disrupt the watermark, the underlying content may also be degraded, undermining the value of invisible watermarking. It would strengthen the paper to include an analysis of this trade-off, perhaps through curves illustrating detection accuracy versus content quality as perturbation levels increase. Additionally, discussing which watermarking methods offer the best trade-off, and identifying which perturbations most effectively disrupt this trade-off, would provide deeper insights.

---

### Review · Reviewer_1nmR · 2024-10-21

**Recommendation:** 3
**Confidence:** 2

**Summary Of Contributions:**

This paper aims to assess the robustness of multimodal watermarks (used in images and text) against real-world perturbations. The study tested various watermarking methods under multiple types of image and text corruptions. The findings suggest vulnerabilities in multimodal watermarking, with some methods being more resilient than others.

**Strengths:**

This work is well written. Extensive experiments are conducted to analyze the robustness of multimodal watermarking. I think it would bring the research community new insight and benefit the future researches in this filed.

**Audience:**

Yes

**Broader Impact Concerns:**

A section of ethical implications is suggested to add to discuss how the robustness of multimodal watermarking will affect the real-world applications and the potential misuse.

**Claims And Evidence:**

This paper introduces multi-modal watermarking benchmark. All details, including perturbation methods, watermarking methods, benchmark models and evaluation metrics are well introduced.

**Datasets And Benchmarks:**

The reproducibility is good.

**Extended Submissions:**

N/A

**Limitations:**

Please refer to the weakness.

**Requested Changes:**

Based on the weakness, I think the authors are suggested to highlight the **multi-modality**. Current draft ignores this point and mainly analyzes the multimodal watermarking from image and text perspectives, respectively. More details can refer to the weakness.

**Strengths And Weaknesses:**

### Strengths
1. The idea is straightforward, and the writing is good.
2. Extensive experiments are conducted.
3. The analysis and discussions benefit future researches.
4. Source code is available.
### Weakness
1. The single-modality perturbations are considered in the experiments. I think it would be better to include more multimodal perturbation methods [3].
2. One important watermarking category. i.e., backdoor-based watermarking [1] is missing.
3. Some recent papers [2] study the multi-modality watermarking methodology for pre-trained VLMs. The authors are suggested to involved them into analysis.
[1] Turning Your Weakness Into a Strength: Watermarking Deep Neural Networks by Backdooring.
[2] Watermarking Vision-Language Pre-trained Models for Multi-modal Embedding as a Service.
[3] Transferable Multimodal Attack on Vision-Language Pre-training Models.